# SVD-LLM: Truncation-aware Singular Value Decomposition for Large Language Model Compression

**Xin Wang**[1]  **Yu Zheng**[2]  **Zhongwei Wan**[1]  **Mi Zhang**[1]
[1]The Ohio State University   [2]Michigan State University
`https://github.com/AIoT-MLSys-Lab/SVD-LLM`

## ABSTRACT

The advancements in Large Language Models (LLMs) have been hindered by their substantial sizes, which necessitates LLM compression methods for practical deployment. Singular Value Decomposition (SVD) offers a promising solution for LLM compression. However, state-of-the-art SVD-based LLM compression methods have two key limitations: truncating smaller singular values may lead to higher compression loss, and the lack of update on the compressed weights after SVD truncation. In this work, we propose `SVD-LLM`, a SVD-based post-training LLM compression method that addresses the limitations of existing methods. `SVD-LLM` incorporates a truncation-aware data whitening technique to ensure a direct mapping between singular values and compression loss. Moreover, `SVD-LLM` adopts a parameter update with sequential low-rank approximation to compensate for the accuracy degradation after SVD compression. We evaluate `SVD-LLM` on 10 datasets and seven models from three different LLM families at three different scales. Our results demonstrate the superiority of `SVD-LLM` over state-of-the-arts, especially at high model compression ratios.

## 1 INTRODUCTION

Large Language Models (LLMs) have demonstrated remarkable capabilities in a wide range of tasks such as natural language understanding and generation (Zhao et al., 2023; Gozalo-Brizuela and Garrido-Merchán, 2023). Despite such capabilities, the democratization of LLMs is primarily restricted by their substantial resource demands, which motivates the design of LLM compression methods (Wan et al., 2024a; Wang et al., 2024; Zhu et al., 2024; Zhou et al., 2024). These methods are often performed in a post-training manner without requiring retraining from scratch. Post-training LLM compression methods based on quantization (Yuan et al., 2024; Huang et al., 2024), unstructured pruning (Frantar and Alistarh, 2023), and structured pruning (Ma et al., 2023; Ashkboos et al., 2024; Zhong et al., 2024) have been intensively studied. Despite their success, these methods have certain limitations, such as dependence on specific hardware and low inference speedup. In contrast, compression methods based on low-rank approximation, such as Singular Value Decomposition (SVD) are not limited by those constraints. Moreover, the KV cache of LLMs compressed via SVD at runtime can also be reduced.

Despite these advantages, the potential of SVD for LLM compression has not been fully explored. Several SVD-based LLM compression methods, such as FWSVD (Hsu et al., 2022) and ASVD (Yuan et al., 2023) have been proposed. However, these methods exhibit severe performance degradation when model compression ratio[1] increases. Such limitation can be attributed to two fundamental issues involved in their approaches. ❶ **Misalignment between SVD truncation and compression loss**: both FWSVD and ASVD fail to establish a direct relationship between singular values and model compression loss. As a consequence, truncating smaller singular values in SVD could lead to higher compression loss. ❷ **Lack of model parameter update after SVD truncation**: as model compression ratio increases, the number of singular values that need to be truncated in SVD increases as well. To compensate for the accuracy degradation caused by truncating a larger number of

---

[1]Model compression ratio refers to the percentage of parameter reduction achieved through compression.

singular values, it becomes necessary to update the remaining parameters of the compressed model. Unfortunately, existing SVD-based LLM compression methods do not incorporate such update in their design, and thus fail to compensate for the accuracy degradation especially under high model compression ratios.

In this paper, we propose a SVD-based post-training LLM compression method, `SVD-LLM`, which effectively addresses the two fundamental issues of existing SVD-based LLM compression methods. `SVD-LLM` differs from them in two key aspects. ❶ **Truncation-Aware Data Whitening:** supported by theoretical proof, `SVD-LLM` incorporates a truncation-aware data whitening technique that ensures a *direct mapping* between singular values and model compression loss. In doing so, the proposed truncation-aware data whitening technique is able to identify which singular values should be truncated to incur minimal model compression loss. ❷ **Parameter Update with Sequential Low-rank Approximation:** to compensate for accuracy degradation after compression, `SVD-LLM` sequentially fine-tunes the decomposed low-ranking matrices for a global accuracy recovery.

We compare `SVD-LLM` with both state-of-the-art SVD-based LLM compression methods as well as pruning and quantization-based LLM compression methods. To demonstrate the generability of `SVD-LLM`, we conduct our evaluation on a total of 10 datasets and seven models from three different LLM families (LLaMA, OPT, and Mistral) at three different scales (7B, 13B, 30B), and evaluate the performance of `SVD-LLM` on both GPU and CPU. We highlight three of our findings:

- `SVD-LLM` outperforms state-of-the-art SVD-based LLM compression methods FWSVD and ASVD across all 10 datasets, three LLM families at three scales by a large margin.
- `SVD-LLM` also outperforms state-of-the-art structured pruning-based LLM compression methods, including LLM-Pruner, SliceGPT, BlockPruner as well as state-of-the-art 1-bit post-training quantization-based LLM compression methods, including PB-LLM and BiLLM. More importantly, when combined with 2-bit post-training quantization, `SVD-LLM` outperforms state-of-the-art 1-bit training-required quantization-based LLM compression method OneBit, presenting a new way to achieve state-of-the-art compression performance without incurring expensive retraining.
- LLMs compressed by `SVD-LLM` are able to achieve inference speedup and memory reduction when deployed on real hardware, including both GPU and CPU. At the same time, `SVD-LLM` is able to reduce runtime KV cache memory without additional accuracy drop.

## 2 RELATED WORK

**Large Language Model Compression:** LLMs in general contain billion-scale parameters. Applying conventional model compression methods for LLMs is unfeasible as they necessitate resource-intensive retraining. Given that, post-training methods that avoid retraining in the compression process have been proposed. In general, these methods can be grouped into four categories: unstructured pruning, structured pruning, quantization, and low-rank approximation. Specifically, unstructured pruning methods (Frantar and Alistarh, 2023) set the individual weights of an LLM to zero without changing its shape. However, irregular sparsification of unstructured pruning is difficult to achieve the desired speedup or memory saving. Unlike unstructured pruning, structured pruning methods (Ma et al., 2023; Ashkboos et al., 2024; Zhong et al., 2024) remove entire channels or other structured components from LLMs, making them easier to implement on hardware. One notable contribution is LLM-Pruner (Ma et al., 2023), which groups weight matrices based on their dependency and assigns the pruning ratio to each group based on the estimated importance. Quantization methods (Lin et al., 2024) compress models by reducing the precision of weight matrices of the LLM. However, similar to unstructured pruning, quantization is also difficult to achieve the desired inference speedup due to the lack of hardware support and efficient kernels for low-precision computation (Lin et al., 2024). Recent studies including PB-LLM (Yuan et al., 2024) and BiLLM (Huang et al., 2024) push the frontier to 1-bit quantization. Nevertheless, these approaches often lead to severe accuracy degradation.

**SVD for Language Model Compression:** Singular Value Decomposition (SVD) is a widely used low-rank approximation technique to reduce matrix size by approximating a matrix with two smaller low-ranking matrices (Golub et al., 1987). Given that, SVD is commonly used for model compression. For instance, DRONE (Chen et al., 2021) achieves optimal SVD compression for small language models such as BERT. However, during SVD compression, DRONE caches all the input activations, making it challenging for LLM compression due to excessive memory usage. For LLMs, directly

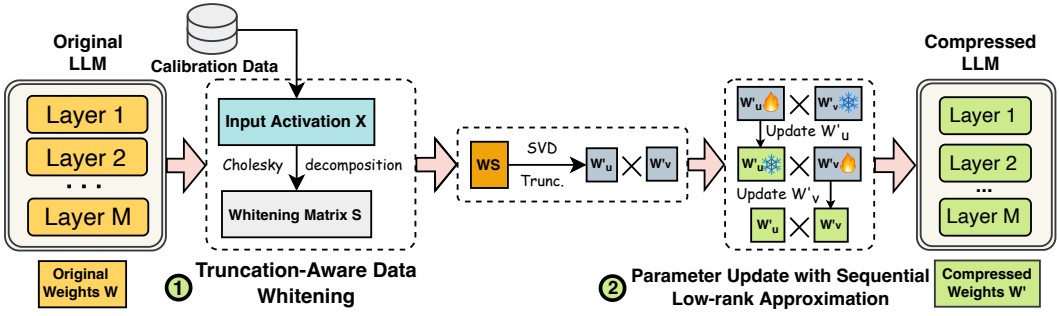

Figure 1: Overview of `SVD-LLM`.

applying SVD on the weight matrix without considering the importance of the weights leads to a large compression loss. To address this issue, Hsu et al. (2022) propose FWSVD, which introduces Fisher information to weigh the importance of parameters. However, FWSVD requires a complex gradient calculation that demands substantial computing and memory resources for LLM compression. Another issue of directly applying SVD is that the distribution of activation can affect the compression loss. To address it, Yuan et al. (2023) propose ASVD, which scales the weight matrix by a diagonal matrix that normalizes the impact of input channels on the weights. However, both FWSVD and ASVD do not establish a direct relationship between singular values and compression loss. As a consequence, truncating the smaller singular values may lead to higher compression loss. Moreover, as compression ratio increases, it becomes necessary to update the compressed weights for accuracy recovery. However, both FWSVD and ASVD do not take it into consideration, and thus incur severe accuracy degradation under high compression ratios.

## 3 SVD-LLM

Figure 1 provides an overview of `SVD-LLM`. At a high level, `SVD-LLM` is a SVD-based post-training LLM compression method. Specifically, following the standard procedure of post-training LLM compression methods (Frantar and Alistarh, 2023; Yuan et al., 2023; Xiao et al., 2023), `SVD-LLM` uses a random set of sentences as calibration data to generate activation for truncation-aware data whitening. Given the generated activation, `SVD-LLM` derives the whitening matrix $S$ through Cholesky decomposition, and then performs SVD to truncate the multiplication of weight matrices $W$ and whitening matrix $S$ to compress the LLM. After truncation, `SVD-LLM` updates the remaining model parameters with sequential low-rank approximation to recover accuracy. In the following, we describe both truncation-aware data whitening and parameter update with sequential low-rank approximation in detail. The pseudocode of `SVD-LLM` is provided in Appendix A.1.

### 3.1 TRUNCATION-AWARE DATA WHITENING

**Motivation:** Due to high variance of input activation, simply applying vanilla SVD for LLM compression leads to severe accuracy degradation (Yuan et al., 2023). To address this issue, existing methods (Yuan et al., 2023; Hsu et al., 2022) formulate LLM compression as an optimization problem with the following objective:

$$O = \min(||WX - W'X||_F) \tag{1}$$

where $W$ is the weight matrix of the original LLM, $X$ is the activation of $W$, $W'$ is the compressed weight matrix, and $||WX - W'X||_F$ is the compression loss in the form of Frobenius loss.

Although existing methods attempt to reduce this compression loss during their SVD truncation, they all fail to establish a direct relationship between singular values and compression loss. As a consequence, truncating smaller singular values in SVD could lead to significant compression loss. Taking ASVD (Yuan et al., 2023) as an example, ASVD extracts a diagonal matrix $S_0$ from $X$ where each element in the diagonal is the absolute mean value of each channel. It then uses $S_0$ to normalize $X$ and converts $WX$ into $(WS_0)(S_0^{-1}X)$. Subsequently, SVD is performed on $WS_0$ to obtain the decomposed matrices $U_0$, $\Sigma_0$, and $V_0$. Lastly, ASVD truncates the smallest singular values in $\Sigma_0$ to obtain the compressed weight matrix $W'_0 = U_0 \times \text{Trunc.}(\Sigma_0) \times V_0 \times S_0^{-1}$.

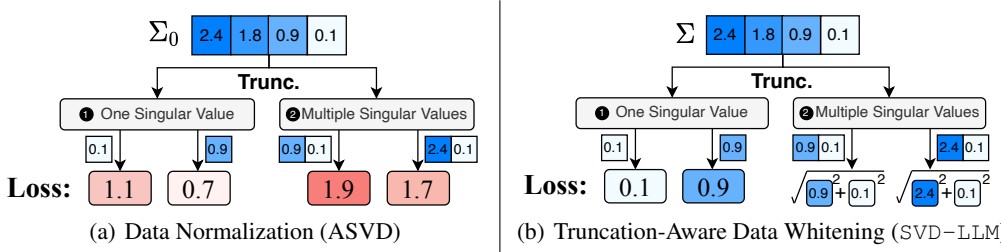

Figure 2: Compression loss ($L = ||WX - W'X||_F$) of different data preprocessing methods.

Although normalizing the activation improves performance, ASVD does not establish a direct relationship between singular values and compression loss (a detailed proof is included in Appendix A.2). To better illustrate this point, we show two concrete examples in Figure 2(a). In example ❶ where only one singular value is truncated, truncating the smallest singular value 0.1 results in a higher compression loss (loss = 1.1) compared to truncating the second smallest singular value 0.9 (loss = 0.7). In example ❷ where multiple singular values are truncated, truncating the smallest two singular values 0.9 and 0.1 also leads to a higher loss (loss = 1.9) than truncating 2.4 and 0.1 (loss = 1.7). As such, truncating the smallest singular values does not lead to minimal loss.

**Key Design:** The key idea of SVD-LLM is to incorporate a truncation-aware data whitening technique that ensures a *direct* mapping between singular values and compression loss. To achieve this, SVD-LLM enforces the whitened activation $S^{-1}X$ to be orthonormal such that each channel is independent of each other, i.e., $(S^{-1}X)(S^{-1}X)^T = S^{-1}XX^T(S^{-1})^T = I$, where $S$ is derived through Cholesky decomposition (Meyer, 2000). SVD is then performed on $WS$ to obtain the decomposed matrices $U, \Sigma, V$, where $U = [u_1, u_2, u_3, ..., u_r]$, $\Sigma = \text{diag}(\sigma_1, \sigma_2, \sigma_3, \cdots, \sigma_r)$, and $V = [v_1, v_2, v_3, ..., v_r]$. Lastly, the smallest singular values in $\Sigma$ are truncated (denoted by $\textbf{Trunc.}(\Sigma)$) to obtain two low-ranking matrices $W'_u = U \times [\textbf{Trunc.}(\Sigma)]^{\frac{1}{2}}, W'_v = [\textbf{Trunc.}(\Sigma)]^{\frac{1}{2}} \times V^T \times S^{-1}$ and the compressed weight matrix $W' = W'_u \times W'_v = U \times \textbf{Trunc.}(\Sigma) \times V^T \times S^{-1}$.

Figure 2(b) illustrates the effect of the proposed truncation-aware data whitening method. In example ❶ where only one singular value is truncated, the compression loss equals to the truncated singular value. In example ❷, the compression loss of truncating multiple singular values equals to the square root of the sum of their squares. As such, under the proposed truncation-aware data whitening technique, truncating the smallest singular values leads to minimal compression loss.

In the following, we provide a theoretical proof on why the proposed truncation-aware data whitening technique ensures a direct mapping between singular values and compression loss in the case of one singular value (Theorem 3.2) and multiple singular values (Corollary 3.3), respectively.

**Lemma 3.1.** *The Frobenius norm of matrix $A$ with dimension $m \times n$ can be deduced into the square root of the trace of its gram matrix, which is:*

$$\|A\|_F \triangleq \left( \sum_{j=1}^{n} \sum_{i=1}^{m} |a_{ij}|^2 \right)^{\frac{1}{2}} = \left[ \textbf{Trace} \left( A^T A \right) \right]^{\frac{1}{2}} \quad (2)$$

Let $\textbf{SVD}(WS)$ denote SVD compression on matrix $WS$. The compressed weight matrix $W'$ can be expressed as $W' = \textbf{SVD}(WS)S^{-1}$. Using Lemma 3.1, we obtain the compression loss $L_i$ when truncating the $i^{th}$ singular value of $WS$ to reduce its rank for compression:

$$L_i = \|WX - W'X\|_F = \left\|WSS^{-1}X - \textbf{SVD}(WS)S^{-1}X\right\|_F = \left\|(WS - \textbf{SVD}(WS))S^{-1}X\right\|_F$$

$$= \left\|\sigma_i u_i v_i^T S^{-1}X\right\|_F = \sigma_i \textbf{Trace} \left( u_i v_i^T S^{-1}XX^T \left(S^{-1}\right)^T v_i u_i^T \right)^{\frac{1}{2}}$$

$$(3)$$

Since both $U = [u_1, u_2, u_3, ..., u_r]$ and $V = [v_1, v_2, v_3, ..., v_r]$ are orthonormal matrices, we have:

$$v_i^T v_i = u_i^T u_i = 1; v_i^T v_j = u_i^T u_j = 0, \forall i \neq j; \textbf{Trace}(v_i v_i^T) = \textbf{Trace}(u_i u_i^T) = 1 \quad (4)$$

**Theorem 3.2.** *If $S$ is the Cholesky decomposition of $XX^T$, the compression loss $L_i$ equals to $\sigma_i$.*

*Proof.* Since the whitening matrix $S$ is the Cholesky decomposition of $XX^T$, we have $SS^T = XX^T$. We can further infer Equation (3) to obtain:

$$L_i = \sigma_i \mathbf{Trace}(u_i v_i^T v_i u_i^T)^{\frac{1}{2}} = \sigma_i \mathbf{Trace}\left(u_i \left(v_i^T v_i\right) u_i^T\right)^{\frac{1}{2}} = \sigma_i \mathbf{Trace}\left(u_i u_i^T\right)^{\frac{1}{2}} = \sigma_i \quad (5)$$

Therefore, $L_i$ of truncating $\sigma_i$ equals to the singular value $\sigma_i$ itself. $\square$

**Corollary 3.3.** *If $S$ is the Cholesky decomposition of $XX^T$, truncating the smallest singular values leads to the lowest loss $L$ compared to truncating others.*

*Proof.* If we truncate $\sigma_{m+1}, \sigma_{m+2}, \sigma_{m+3}, ..., \sigma_r$ in $\Sigma$ for compression, the square of the loss $L$ is:

$$
\begin{aligned}
L^2 &= \left\| \sum_{i=m+1}^{r} \sigma_i u_i v_i^T S^{-1} X \right\|_F^2 = \sum_{j=m+1}^{r} \sum_{i=m+1}^{r} \sigma_i \sigma_j \mathbf{Trace}(u_i v_i^T S^{-1} XX^T (S^{-1})^T v_j u_j^T) \\
&= \sum_{i=m+1}^{r} \sigma_i^2 \mathbf{Trace}(u_i v_i^T S^{-1} XX^T (S^{-1})^T v_i u_i^T) = \sum_{i=m+1}^{r} (L_i)^2 = \sum_{i=m+1}^{r} (\sigma_i)^2
\end{aligned}
\quad (6)
$$

The squared loss $L^2$ equals to the sum of the squared singular values (More detailed derivation is in Appendix A.3). Truncating the smallest singular values achieves the lowest compression loss. $\square$

Given that our proposed truncation-aware data whitening technique is built upon SVD, whose applicability depends on certain singular value distribution, we further conduct a spectrum analysis of the singular values obtained by our technique. We refer the readers to Appendix A.4 for details.

It should also be noted that our proposed truncation-aware data whitening technique not only ensures a direct mapping between singular values and compression loss, but is also able to achieve the same theoretical optimal compression loss as DRONE (Chen et al., 2021). However, during SVD compression, DRONE stores all input activations in memory, which poses a challenge for LLM compression due to high memory consumption. In contrast, `SVD-LLM` incrementally updates its $XX^T$ matrix by adding the $xx^T$ of each new input $x$, which is considerably smaller than the full input activation. In Appendix A.5, we provide our proof on `SVD-LLM` achieving the same theoretical optimal compression loss as DRONE and discuss the advantages of `SVD-LLM` over DRONE in memory saving, compression speed, and numerical stability in details.

### 3.2 Parameter Update with Sequential Low-rank Approximation

**Motivation:** Although the proposed truncation-aware data whitening technique helps preserve the accuracy during compression, as the compression ratio increases, the accuracy of the compressed LLM may degrade given that more larger singular values will be truncated by SVD compression. Therefore, it is necessary to update the remaining parameters in the compressed LLM.

**Key Design:** `SVD-LLM` proposes a variant of LoRA fine-tuning to update the remaining weight parameters of the compressed LLM for accuracy recovery. Specifically, given the two low-ranking matrices $W_u', W_v'$ generated by truncation-aware data whitening, instead of directly applying LoRA fine-tuning to the compressed weight matrix $W' = W_u' \times W_v'$ as standard LoRA does, we propose to apply LoRA on top of $W_u'$ and $W_v'$ separately to preserve their low-rank structures as follows:

$$W_u' \leftarrow W_u' + B_u A_u, W_v' \leftarrow W_v' + B_v A_v \quad (7)$$

where $A_u$, $B_u$, $A_v$, and $B_v$ are matrices used for LoRA fine-tuning.

Simultaneously fine-tuning $W_u'$ and $W_v'$ will not guarantee a decrease in fine-tuning loss. This is because the derivatives of $W_u'$ and $W_v'$ are interdependent during the fine-tuning process, where optimization of one matrix may interfere with the optimization of the other. Therefore, as shown in Figure 1, we propose a sequential low-rank approximation strategy to fine-tune $W_u'$ and $W_v'$ in a sequential manner. Specifically, we first freeze matrix $W_v'$ and fine-tune $W_u'$ with LoRA. We then perform the second-round LoRA fine-tuning on matrix $W_v'$ while freezing the updated weight matrix $W_u'$. Finally, we add $B_u \times A_u$ and $B_v \times A_v$ matrices to $W_u'$ and $W_v'$ to obtain the final compressed weight matrices.

Table 1: Performance of LLaMA-7B compressed by `SVD-LLM`, `SVD-LLM (W)`, and baselines under different compression ratio (corresponding weight memory is listed inside bracket) on two language modeling datasets (measured by perplexity (↓)), eight common sense reasoning datasets (six measured by both individual and average accuracy (↑), TruthfulQA measured by BLEU score (↑), and GSM8K measured by Exact Match Accuracy (↑)). The best performance is marked in bold. The relative performance gain compared to the best-performing baseline is marked in green inside bracket.

| Ratio (Mem.) | Method | WikiText-2↓ | C4↓ | Openb. | ARC_e | WinoG. | HellaS. | PIQA | MathQA | Average↑ | TruthfulQA↑ | GSM8K↑ |
|---|---|---|---|---|---|---|---|---|---|---|---|---|
| 0% (13.5 GB) | Original | 5.68 | 7.34 | 0.34 | 0.75 | 0.70 | 0.57 | 0.79 | 0.27 | 0.57 | 0.30 | 0.09 |
| 20% (10.2 GB) | SVD | 20061 | 18800 | 0.05 | 0.04 | 0.01 | 0.03 | 0.02 | 0.03 | 0.03 | 0.00 | 0.00 |
| | FWSVD | 1727 | 1511 | 0.09 | 0.11 | 0.05 | 0.08 | 0.10 | 0.05 | 0.08 | 0.00 | 0.00 |
| | ASVD | 11.14 | 15.93 | 0.29 | 0.53 | 0.64 | 0.41 | 0.68 | 0.17 | 0.45 | 0.21 | 0.04 |
| | SVD-LLM (W) | 7.94 (↓29%) | 15.84 (↓1%) | 0.31 | 0.62 | 0.61 | 0.45 | 0.71 | 0.21 | 0.49 (↑9%) | 0.26 (+0.05) | 0.05 (+0.01) |
| | SVD-LLM | **7.73** (↓31%) | **12.23** (↓23%) | **0.33** | **0.67** | **0.69** | **0.55** | **0.79** | **0.26** | **0.55** (↑22%) | **0.28** (+0.07) | **0.08** (+0.04) |
| 40% (7.76 GB) | SVD | 52489 | 47774 | 0.04 | 0.04 | 0.05 | 0.01 | 0.03 | 0.02 | 0.03 | 0.00 | 0.00 |
| | FWSVD | 18156 | 12847 | 0.06 | 0.05 | 0.02 | 0.00 | 0.05 | 0.03 | 0.04 | 0.00 | 0.00 |
| | ASVD | 1407 | 1109 | 0.08 | 0.11 | 0.09 | 0.08 | 0.13 | 0.08 | 0.10 | 0.01 | 0.00 |
| | SVD-LLM (W) | 13.73 (↓99%) | 75.42 (↓93%) | 0.25 | 0.33 | 0.55 | 0.40 | 0.63 | 0.12 | 0.38 (↑280%) | 0.17 (+0.17) | 0.02 (+0.02) |
| | SVD-LLM | **9.27** (↓99%) | **15.63** (↓99%) | **0.29** | **0.59** | **0.68** | **0.52** | **0.69** | **0.20** | **0.50** (↑400%) | **0.24** (+0.23) | **0.07** (+0.07) |
| 60% (5.35 GB) | SVD | 105474 | 106976 | 0.01 | 0.03 | 0.01 | 0.00 | 0.01 | 0.02 | 0.01 | 0.00 | 0.00 |
| | FWSVD | 32194 | 29292 | 0.06 | 0.02 | 0.01 | 0.01 | 0.02 | 0.03 | 0.03 | 0.00 | 0.00 |
| | ASVD | 57057 | 43036 | 0.05 | 0.04 | 0.06 | 0.09 | 0.08 | 0.05 | 0.06 | 0.00 | 0.00 |
| | SVD-LLM (W) | 66.62 (↓99%) | 471.83 (↓99%) | 0.10 | 0.05 | 0.17 | 0.10 | 0.21 | 0.04 | 0.11 (↑83%) | 0.01 (+0.01) | 0.00 (+0.00) |
| | SVD-LLM | **15.00** (↓99%) | **26.26** (↓99%) | **0.18** | **0.42** | **0.44** | **0.31** | **0.35** | **0.12** | **0.30** (↑400%) | **0.14** (+0.14) | **0.04** (+0.04) |
| 80% (2.58 GB) | SVD | 687291 | 708243 | 0.00 | 0.01 | 0.02 | 0.01 | 0.01 | 0.00 | 0.01 | 0.00 | 0.00 |
| | FWSVD | 96872 | 89243 | 0.01 | 0.02 | 0.00 | 0.01 | 0.01 | 0.00 | 0.01 | 0.00 | 0.00 |
| | ASVD | 80425 | 67927 | 0.04 | 0.03 | 0.03 | 0.02 | 0.01 | 0.01 | 0.03 | 0.00 | 0.00 |
| | SVD-LLM (W) | 1349 (↓98%) | 6224 (↓91%) | 0.07 | 0.03 | 0.04 | 0.02 | 0.07 | 0.01 | 0.04 (↑33%) | 0.00 (+0.00) | 0.00 (+0.00) |
| | SVD-LLM | **31.79** (↓99%) | **43.71** (↓99%) | **0.11** | **0.23** | **0.21** | **0.14** | **0.17** | **0.08** | **0.16** (↑433%) | **0.04** (+0.04) | **0.02** (+0.02) |

# 4 Experiments and Analysis

**Baselines.** We compare `SVD-LLM` against three groups of methods: (1) Vanilla SVD and state-of-the-art SVD-based LLM compression methods: FWSVD (Hsu et al., 2022), ASVD (Yuan et al., 2023) (Section 4.1) and FLAP (Appendix A.6). (2) Other types of LLM compression methods. These include three state-of-the-art pruning-based LLM compression methods: LLM-Pruner (Ma et al., 2023), SliceGPT (Ashkboos et al., 2024), and BlockPruner (Zhong et al., 2024), and three state-of-the-art quantization-based LLM compression methods: PB-LLM (Yuan et al., 2024), BiLLM (Huang et al., 2024), and OneBit (Xu et al., 2024) (Section 4.4). (3) Smaller LLM StableLM-3B (Tow et al.) pre-trained from scratch (Appendix A.7).

**Models and Datasets.** To demonstrate the generability of `SVD-LLM`, we evaluate the performance of `SVD-LLM` on seven models from three different LLM families at three different scales (LLaMA-7B, 13B, 30B, LLaMA2-7B (Touvron et al., 2023), OPT-6.7B (Zhang et al., 2022), Vicuna-7B (Chiang et al., 2023) and Mistral-7B (Jiang et al., 2023)) and 10 datasets including two language modeling datasets (WikiText-2 (Merity et al., 2017), and C4 (Raffel et al., 2020)), six classification datasets (OpenbookQA (Mihaylov et al., 2018), WinoGrande (Sakaguchi et al., 2020), HellaSwag (Zellers et al., 2019), Arc_e (Clark et al., 2018), PIQA (Bisk et al., 2020), MathQA (Amini et al., 2019)), and two generation datasets (TruthfulQA (Lin et al., 2022) and GSM8K (Cobbe et al., 2021)) with the LM-Evaluation-Harness framework (Gao et al., 2023).

**Implementation Details.** To ensure a fair comparison, we followed ASVD (Yuan et al., 2023) to randomly select 256 samples from WikiText-2 as the calibration data. We followed the same configuration used in LLM-Pruner (Ma et al., 2023) to use Alpaca (Taori et al., 2023) dataset with 50K samples for parameter update in `SVD-LLM`. The inference efficiency experiment is conducted on both NVIDIA A100 GPU and AMD EPYC 7643 CPU while the other experiments are conducted on NVIDIA A100 GPUs.

## 4.1 Comparison with State-of-the-Art SVD-based LLM compression Methods

First, we compare the performance of `SVD-LLM` with state-of-the-art SVD-based LLM compression methods from four aspects: (1) performance under different compression ratios, (2) performance on different LLMs, (3) performance on LLMs with larger scales, and (4) compression speed (Appendix A.8). Given that all the SVD-based baselines do not incorporate LoRa fine-tuning, to ensure a fair comparison, we also compare to `SVD-LLM` with truncation-aware data whitening only (denoted as `SVD-LLM (W)`).

**Performance under Different Compression Ratios.** We first evaluate the performance of LLaMA-7B compressed by `SVD-LLM`, `SVD-LLM (W)`, and the SVD-based baselines under compression

Table 2: Perplexity ($\downarrow$) of `SVD-LLM`, `SVD-LLM (W)`, and baselines on WikiText-2 and the average accuracy ($\uparrow$) of the six common sense reasoning datasets of four different LLMs – OPT-6.7B, LLaMA 2-7B, Mistral-7B, and Vicuna-7B – under 20% compression ratio. The relative performance gain compared to the best-performing baseline is marked in green color inside bracket.

| METHOD | OPT-6.7B | | LLaMA 2-7B | | MISTRAL-7B | | VICUNA-7B | |
|---|---|---|---|---|---|---|---|---|
| | Perplexity$\downarrow$ | Accuracy$\uparrow$ | Perplexity$\downarrow$ | Accuracy$\uparrow$ | Perplexity$\downarrow$ | Accuracy$\uparrow$ | Perplexity$\downarrow$ | Accuracy$\uparrow$ |
| Original | 10.86 | 0.52 | 5.47 | 0.57 | 5.25 | 0.61 | 6.78 | 0.56 |
| SVD | 66275 | 0.03 | 18192 | 0.09 | 159627 | 0.03 | 18644 | 0.05 |
| FWSVD | 14559 | 0.06 | 2360 | 0.12 | 6357 | 0.08 | 2758 | 0.09 |
| ASVD | 82.00 | 0.32 | 10.10 | 0.36 | 13.72 | 0.32 | 16.23 | 0.33 |
| SVD-LLM (W) | 16.04 (↓80%) | 0.41 (↑28%) | 8.50 (↓16%) | 0.53 (↑47%) | 10.21 (↓26%) | 0.42 (↑24%) | 8.41 (↓48%) | 0.51 (↑55%) |
| SVD-LLM | **14.47** (↓82%) | **0.49** (↑53%) | **7.73** (↓23%) | **0.54** (↑50%) | **7.47** (↓45%) | **0.55** (↑72%) | **7.43** (↓54%) | **0.54** (↑64%) |

ratios ranging from 20% to 80% on all 10 datasets. The results are summarized inTable 1. Both `SVD-LLM` and `SVD-LLM (W)` consistently outperform vanilla SVD, FWSVD and ASVD across all the compression ratios. In particular, when the compression ratio is 40% and above, `SVD-LLM` reduces the perplexity by more than 99% on two language modeling datasets and achieves over 400% higher average accuracy on six classification datasets. More importantly, the results on two generation datasets (TruthfulQA, GSM8K) of all three baselines when compression ratios are 60% and above are zero, meaning that the compressed LLMs totally lose their generation ability. In contrast, `SVD-LLM` still outputs good generation even under 80% compression ratio. Example contents generated by the compressed LLMs are included in Appendix A.9.

**Performance on Different LLMs.** To examine the generability of `SVD-LLM` and `SVD-LLM (W)` across different LLMs, we compare the performance between `SVD-LLM` and the baselines on four different models from three different LLM families – OPT-6.7B (OPT family), LLaMA 2-7B (LLaMA family), Mistral-7B (Mistral family), and Vicuna-7B (LLaMA family) – under 20% compression ratio on WikiText-2 and six classification datasets. As shown in Table 2, both `SVD-LLM` and `SVD-LLM (W)` consistently outperform baselines on all four LLMs, and exhibits more stable performance across different LLMs, especially compared to vanilla SVD and FWSVD.

Table 3: Perplexity ($\downarrow$) of `SVD-LLM`, `SVD-LLM (W)`, and baselines on WikiText-2 and the average accuracy ($\uparrow$) of the six classification datasets of LLaMA-13B and LLaMA-30B under 20% compression ratio. The relative performance gain compared to the best-performing baseline is marked in green color inside bracket.

| METHOD | LLaMA-13B | | LLaMA-30B | |
|---|---|---|---|---|
| | Perplexity$\downarrow$ | Accuracy$\uparrow$ | Perplexity$\downarrow$ | Accuracy$\uparrow$ |
| Original | 5.09 | 0.59 | 4.10 | 0.61 |
| SVD | 946.31 | 0.21 | 54.11 | 0.33 |
| FWSVD | 15.98 | 0.43 | 20.54 | 0.42 |
| ASVD | 6.74 | 0.54 | 22.71 | 0.44 |
| SVD-LLM (W) | 6.61 (↓2%) | 0.54 (↑0%) | 5.63 (↓73%) | 0.57 (↑30%) |
| SVD-LLM | **6.43** (↓5%) | **0.55** (↑2%) | **5.14** (↓75%) | **0.59** (↑34%) |

**Performance on LLMs with Larger Scales.** To examine the generability of `SVD-LLM` and `SVD-LLM (W)` on LLMs with larger scales, we compare the performance between `SVD-LLM` and the baselines on LLaMA-13B and LLaMA-30B under 20% compression ratio. As shown in Table 3, both `SVD-LLM` and `SVD-LLM (W)` consistently outperform vanilla SVD, FWSVD, and ASVD on both model sizes.

## 4.2 INFERENCE EFFICIENCY OF `SVD-LLM`

**Theoretical Analysis of Inference Efficiency.** Assume `SVD-LLM` compresses the weight matrix $W \in \mathbb{R}^{d \times n}$ into two low-ranking matrices $W'_u \in \mathbb{R}^{d \times r}, W'_v \in \mathbb{R}^{r \times n}$. The compression ratio is then calculated as $R_w = 1 - \frac{(d+n)r}{dn}$.

(1) Compute Complexity Analysis: Given input $X \in \mathbb{R}^{n \times d}$, instead of recalculating the full weight matrix $W' = W'_u \times W'_v$ and then computing the output $W' \times X$, `SVD-LLM` calculates the intermediate state $M = W'_v \times X$ and then computes the output $Y = W'_u \times M$. In this way, the computation complexity is reduced from $O\left(d^2 n\right)$ to $O\left(d^2 r + rnd\right)$. Taking compression ratio $R_w = 50\%$ as an example, since $R_w = 1 - \frac{(d+n)r}{dn}$, we have $r = \frac{dn}{2(d+n)}$. Then the computation complexity is $O\left(d^2 r + rnd\right) = O(rd(d+n)) = O\left(\frac{d^2 n}{2}\right) = \frac{1}{2}O\left(d^2 n\right)$, which reduces 50%. In general, given any compression ratio $R_w$, the computation complexity is reduced to $1 - R_w$ times of the original.

(2) Inference Memory Analysis: Since `SVD-LLM` does not recalculate the full weight $W' = W'_u \times W'_v$, the weight memory is reduced to $1 - R_w$ times of the original during inference. As another

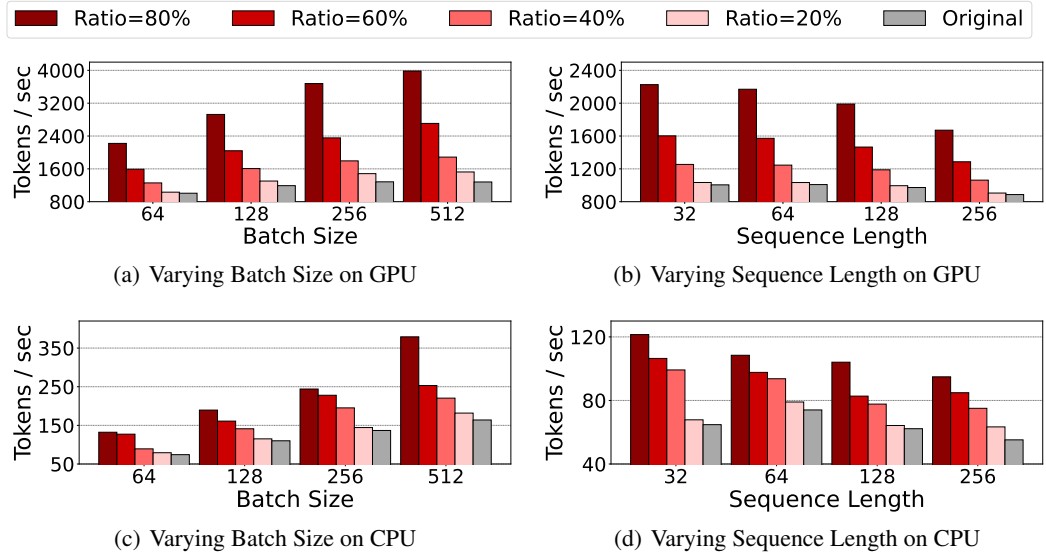

(a) Varying Batch Size on GPU       (b) Varying Sequence Length on GPU

(c) Varying Batch Size on CPU       (d) Varying Sequence Length on CPU

Figure 3: Throughput (tokens/sec) of LLaMA-7B and its compressed version by `SVD-LLM` under different compression ratios on a single A100 GPU under different batch sizes (a) and different sequence lengths (b) and on a single AMD EPYC 7643 CPU under different batch sizes (c) and different sequence lengths (d). For (a) and (c), sequence length is 32. For (b) and (d), batch size is 64.

advantage, `SVD-LLM` is able to reduce the runtime KV cache memory as well (Wan et al., 2024b; 2025; Shen et al., 2024). Specifically, instead of keeping $W'_u \times W'_v \times X$ in the KV cache, `SVD-LLM` provides the option to store the intermediate result $M = W'_v \times X$ in the KV cache and recomputes the original key and value states with the decomposed weight matrix $W'_u$ if required. As such, the runtime KV cache is reduced to $\frac{r}{d} = (1 - R_w) \times \frac{d}{n+d}$ times of the original. Moreover, since $W'_u$ is already stored as the weight matrix in the decomposed LLM, the original intermediate state matrix can still be recovered by $W'_u \times M$ without accuracy drop. Therefore, `SVD-LLM` provides a unified solution that enables *simultaneous* model compression and KV cache compression.

**Inference Speedup on Hardware.** To quantify inference speedup achieved by `SVD-LLM` on real hardware, we measure the numbers of tokens that LLaMA-7B and its compressed version by `SVD-LLM` generate per second (i.e., throughput) under different batch sizes and sequence lengths on a single NVIDIA A100 GPU and a single AMD EPYC 7643 CPU, respectively. The results are shown in Figure 3. We have three observations. (1) Under a specific batch size or sequence length, the speedup achieved by `SVD-LLM` related to the original model increases as the compression ratio increases. (2) Under a specific compression ratio, the speedup achieved by `SVD-LLM` related to the original model becomes more significant as the batch size increases or as the sequence length decreases. (3) The above two observations are valid for both GPU and CPU.

**Inference Memory Reduction on Hardware.** Lastly, we evaluate the inference memory saving, including both weight memory and runtime KV cache memory saving on a single A100 GPU. Specifically, we measure the peak memory footprint during inference when generating 128 tokens with batch size of 32 using LLaMA-7B compressed by `SVD-LLM` under different compression ratios w/ and w/o considering KV cache reduction. The results are illustrated in Figure 4 where the memory reduction from the dotted line to the blue bars comes mainly from model weight compression and the memory reduction from the blue bars to the yellow bars comes mainly from KV cache compression. As shown, both weight memory saving and runtime KV cache memory saving brought by `SVD-LLM` are near linear to the compression ratio.

### 4.3 ABLATION STUDY

**Modular Sensitivity Study.** We conduct ablation studies to evaluate the separate contributions of the two key components (i.e., truncation-aware data whitening and parameter update with sequential low-rank approximation) of `SVD-LLM`. Let `SVD-LLM (W)` denote the version of `SVD-LLM` with truncation-aware data whitening only; `SVD-LLM (U)` denote the version of `SVD-LLM` with normal

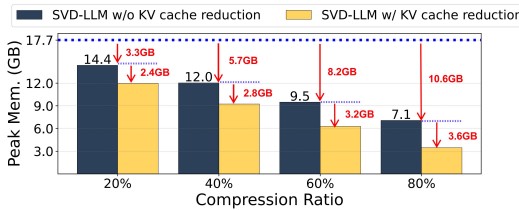

Figure 4: Peak memory to generate 128 tokens with batch size of 32 using LLaMA-7B compressed by SVD-LLM w/ and w/o KV-cache reduction. The dotted line indicates the peak memory of the original LLaMA-7B. The memory reduction from the dotted line to the blue bars mainly comes from the model compression. The memory reduction from the blue to the yellow bars mainly comes from the reduced footprint of the KV cache.

Table 4: Perplexity (↓) of compressed LLaMA-7B on WikiText-2 under different compression ratios. SVD-LLM (W) denotes the version of SVD-LLM with truncation-aware data whitening only; SVD-LLM (U) denotes the version of SVD-LLM with parameter update with sequential low-rank approximation only; The relative performance gain compared to ASVD is marked in green color inside bracket.

| METHOD | 20% | 40% | 60% |
|---|---|---|---|
| ASVD | 11.14 | 1407 | 57057 |
| SVD-LLM (W) | 7.94 (↓29%) | 13.11 (↓99%) | 42.30 (↓99%) |
| SVD-LLM (U) | 10.12 (↓9%) | 19.28 (↓99%) | 49.88 (↓99%) |
| SVD-LLM | **7.73** (↓31%) | **9.27** (↓99%) | **15.00** (↓99%) |

SVD truncation and parameter update with sequential low-rank approximation. As shown in Table 4, we have three observations. (1) SVD-LLM (W), SVD-LLM (U) and SVD-LLM consistently outperform ASVD across all the compression ratios. Notably, when the compression ratio is at and above 40%, all of them reduce the perplexity by more than 99% compared to ASVD. (2) SVD-LLM consistently outperforms both SVD-LLM (U) and SVD-LLM (W) across all compression ratios. This result demonstrates the unique contribution from each of the two key components and the importance of combining both components to achieve the best performance. (3) Comparing between SVD-LLM (W) and SVD-LLM (U), SVD-LLM (W) achieves a lower perplexity compared to SVD-LLM (U) across all compression ratios. This result indicates that truncation-aware data whitening plays a more significant role than parameter update with sequential low-rank approximation.

**Impact of Calibration Data.** Next, we examine the impact of calibration data on SVD-LLM. Figure 5 and Table 6 summarize the performance of compressed LLaMA-7B when changing three key characteristics of the calibration data: (1) number of calibration data, (2) the seed used to randomly sample the calibration data, and (3) dataset from which calibration data is sampled. As shown, the changes of the three key characteristics on calibration data incur no more than 3% to the final performance, indicating that the sensitivity of SVD-LLM on calibration data is limited.

**Impact of Updating Order.** Lastly, we examine the impact of updating order in the parameter update with sequential low-rank approximation component to the final performance of the compressed LLM. Table 5 shows the performance of compressed LLaMA-7B under 20% to 80% compression ratios on WikiText-2 with different updating orders. As shown, there is only a small difference of the final performance between updating matrix $W'_u$ first and updating matrix $W'_v$ first. This result indicates SVD-LLM is not sensitive to the updating order.

More ablation studies are included in Appendix A.10.

## 4.4 COMPARISON WITH OTHER TYPES OF LLM COMPRESSION METHODS

SVD-LLM is orthogonal to other post-training LLM compression methods including pruning and quantization (Wan et al., 2024a; Shen et al., 2025). Lastly, we compare the performance of SVD-LLM with state-of-the-art structured pruning-based and quantization-based LLM compression methods. As discussed in Section 2, since unstructured pruning methods are difficult to achieve its efficiency on hardware, we do not make a comparison with them in this experiment.

**Comparison with Structured Pruning.** First, we compare SVD-LLM with three state-of-the-art structured pruning-based LLM compression methods: LLM-Pruner (Ma et al., 2023), SliceGPT (Ashkboos et al., 2024) and BlockPruner (Zhong et al., 2024) under the same memory budget, ranging from 10 GB to 7 GB on LLaMA-7B using WikiText-2 dataset. As shown in Table 7, SVD-LLM outperforms all three structured pruning-based LLM compression methods. In particular, SVD-LLM achieves up to 56% reduction in perplexity under 7G memory budget.

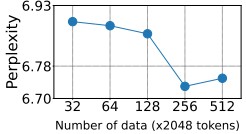 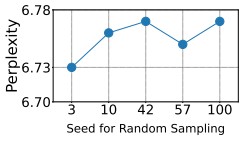

(a) Change of Number  (b) Change of Seed

Figure 5: Perplexity of LLaMA-7B under 20% compression ratio using calibration data sampled with different number or seeds from WikiText-2.

Table 6: Performance of LLaMA-7B compressed by SVD-LLM under 20% compression ratio using calibration data sampled from WikiText-2 (by default in our paper) and C4 datasets. The performance on WikiText-2 and C4 are reported by perplexity (↓), while the performance on six downstream datasets are reported by average accuracy (↑). The performance on TruthfulQA and GSM8K are reported by BLEU score(↑) and Exact Match Accuracy (↑) respectively. The relative performance gain for data sampled from one dataset compared to another is marked in green color inside bracket.

Table 5: Perplexity of LLaMA-7B compressed by SVD-LLM under 20% to 80% compression ratio on WikiText-2 with different updating orders.

| UPDATING ORDER | 20% | 40% | 60% | 80% |
|---|---|---|---|---|
| $W'_u$ first, then $W'_v$ | 7.85 | 9.32 | 13.20 (↓1%) | 31.67(↓1%) |
| $W'_v$ first, then $W'_u$ | 7.73 (↓2%) | 9.27 (↓1%) | 15.00 | 31.79 |

| WikiText-2↓ | C4↓ | Average↑ | TruthfulQA↑ | GSM8K↑ |
|---|---|---|---|---|
| Calibration data sampled from WikiText-2 | | | | |
| 7.73 (↓1%) | 12.23 | 0.55 (↑2%) | 0.28 | 0.08 |
| Calibration data sampled from C4 | | | | |
| 7.79 | 11.97 (↓1%) | 0.54 | 0.28 | 0.08 |

Table 7: Perplexity (↓) of LLaMA-7B compressed by structured pruning methods and SVD-LLM under various memory budget on WikiText-2. The relative performance gain compared to the best-performing baseline is marked in green.

Table 8: Perplexity (↓) of LLaMA-7B compressed by 1-bit quantization methods and SVD-LLM on WikiText-2. The relative performance gain compared to the best-performing baseline is marked in green.

| METHOD | PERPLEXITY UNDER VARIOUS MEMORY BUDGET | | | |
|---|---|---|---|---|
| | 10 GB | 9 GB | 8 GB | 7 GB |
| LLM-Pruner | 9.88 | 12.21 | 18.94 | 21.68 |
| SliceGPT | 8.78 | 12.73 | 16.39 | 27.41 |
| BlockPruner | 9.40 | 12.76 | 19.78 | 43.05 |
| SVD-LLM | 7.92 (↓10%) | 8.18 (↓33%) | 8.33 (↓49%) | 9.63 (↓56%) |

| METHOD | TYPE | MEMORY | PERPLEXITY |
|---|---|---|---|
| PB-LLM | Post-training | 1.9 GB | 104.83 |
| BiLLM | Post-training | 1.5 GB | 47.67 |
| SVD-LLM | Post-training | 1.5 GB | 47.21 (↓1%) |
| OneBit | Training-required | 1.3 GB | 10.20 |
| SVD-LLM (2-bit) | Post-training | 1.3 GB | 9.83 (↓4%) |

**Comparison with Quantization.** Finally, we compare SVD-LLM with three state-of-the-art quantization-based LLM compression methods: BiLLM (Huang et al., 2024), PB-LLM (Yuan et al., 2024), and OneBit (Xu et al., 2024), which push the frontier to 1-bit quantization. Among them, both BiLLM and PB-LLM are post-training methods, and OneBit is training-required. The results on LLaMA-7B using WikiText-2 dataset are shown in Table 8. We have three observations. (1) Compared to post-training methods PB-LLM and BiLLM, SVD-LLM achieves the best performance. (2) Training-required method OneBit outperforms SVD-LLM. This result is expected because OneBit involves resource-intensive retraining using large-scale datasets to boost the accuracy after compression. (3) Lastly, we combine SVD-LLM with a 2-bit quantization-based post-training LLM compression method QuIP# (Tseng et al., 2024). This is achieved by first applying SVD-LLM to the LLM under 40% compression ratio, and then applying QuIP# for 2-bit quantization on the compressed model ($W'_u$ and $W'_v$) generated from SVD-LLM. As shown in Table 8, SVD-LLM outperforms state-of-the-art 1-bit training-required method OneBit *without* involving resource-intensive retraining. This result demonstrates the potential of a hybrid approach – integrating SVD-based and quantization-based compression techniques – to push the boundaries of post-training LLM compression.

## 5 CONCLUSION

In this work, we present SVD-LLM, a SVD-based post-training LLM compression method. SVD-LLM proposes a truncation-aware data whitening technique to guide which singular values to be truncated with minimal compression loss. It also introduces a sequential low-rank approximation strategy to compensate for accuracy degradation caused by singular value truncation. We evaluate SVD-LLM on 10 datasets and seven models from three LLM families at three scales. Our results demonstrate the superiority of SVD-LLM over state-of-the-arts, especially at high model compression ratios.

## 6 ACKNOWLEDGEMENT

This work is supported in part by NSF Award NeTS-2312675.

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

# A APPENDIX.

## A.1 PSEUDOCODE OF SVD-LLM

Algorithm 1 shows the pseudocode of SVD-LLM. Before compression, SVD-LLM randomly collects a small amount of sentences as calibration data $C$, then runs the truncation-aware data whitening process as shown in Algorithm 2 to obtain the set of whitening matrix $\text{Set}_S$ for the weight to compress. After that, it runs the SVD and truncation with $\text{Set}_S$ on each weight matrix in the LLM. Instead of directly finishing the whole compression, it stores the decomposed matrices and further utilizes these matrices to run the parameter update with sequential low-rank approximation as shown in Algorithm 3.

---

**Algorithm 1** Pseudocode of SVD-LLM

---

1: **Input:** $M$: Original LLM
2: **Output:** $M''$: Compressed LLM by SVD-LLM
3: **procedure** SVD-LLM($M$)
4:     Randomly collect several sentences as the calibration data $C$
5:     $\text{Set}_S \leftarrow$ TRUNCATION-AWARE DATA WHITENING($M, C$)
6:     $\text{Set}_W \leftarrow M$                         ▷ Obtain the set of weights in $M$ to compress
7:     **for** $W$ **in** $\text{Set}_W$ **do**
8:         $S \leftarrow \text{Set}_S(W)$            ▷ Extract the whitening matrix of current weight $W$
9:         $U, \Sigma, V \leftarrow \text{SVD}(WS)$        ▷ Apply singular value decomposition on $W$
10:        $\Sigma_1 \leftarrow \text{Trunc.}(\Sigma)$       ▷ Truncate the smallest singular values in $\Sigma$
11:        $W'_u \leftarrow U(\Sigma_1)^{1/2}, W'_v \leftarrow (\Sigma_1)^{1/2}V^T S^{-1}$    ▷ Obtain two low-rank matrices
12:        $M'(W) \leftarrow W'_u, W'_v$       ▷ Replace $W$ with $W'_u$ and $W'_v$ in $L$
13:     **end for**
14:     $M'' \leftarrow$ PARAMETER UPDATE WITH SEQUENTIAL LOW-RANK APPROXIMATION($M'$)
15:     **return** $M''$
16: **end procedure**

---

**Algorithm 2** Pseudocode of Truncation-Aware Data Whitening

---

1: **Input:** $M$: Original LLM
2: **Input:** $C$: Calibration Data
3: **Output:** $\text{Set}_S$: Set of whitening matrices for the weight to compress in $M$
4: **procedure** TRUNCATION-AWARE DATA WHITENING($M, C$)
5:     $\text{Set}_S \leftarrow \emptyset$                      ▷ Initialize the set of whitening matrices
6:     $\text{Set}_W \leftarrow M$                  ▷ Obtain the set of weights in $M$ to compress
7:     **for** $W$ **in** $\text{Set}_W$ **do**
8:         $X \leftarrow M(W, C)$      ▷ Obtain the input activation of the weight matrix $W$
9:         $S \leftarrow \text{Cholesky\_Decomposition}(XX^T)$   ▷ Apply cholesky decomposition on $XX^T$
10:        $\text{Set}_S \leftarrow S \cup \text{Set}_S$      ▷ Store the whitening weight matrix in the set
11:     **end for**
12:     **return** $\text{Set}_S$
13: **end procedure**

---

**Algorithm 3** Pseudocode of Parameter Update with Sequential Low-rank Approximation

---

1: **Input:** $M'$: Compressed LLM by Truncation-aware Data Whitening
2: **Output:** $M''$: Compressed LLM with Parameter Update with Sequential Low-rank Approximation
3: **procedure** PARAMETER UPDATE WITH SEQUENTIAL LOW-RANK APPROXIMATION($M'$)
4:     $M'_u \leftarrow \text{LoRA}_u(M')$           ▷ Fix all $W'_v$, fine-tune all $W'_u$
5:     $M'' \leftarrow \text{LoRA}_v(M'_u)$         ▷ Fix all $W'_u$, fine-tune all $W'_v$
6:     **return** $M''$
7: **end procedure**

---

## A.2 COMPRESSION LOSS OF ASVD

ASVD introduces a diagonal scaling matrix $S_0$ that modifies the weight matrix to reflect the varying significance of different input channels. The linear layer is formulated as $Y = (WS_0)S_0^{-1}X$. The compression is made by keeping the largest $m$ singular value of $WS_0$:

$$WS_0 \approx \sum_{i=1}^{m} \sigma'_i u'_i {v'}_i^T$$

The resulting activation is expressed as:

$$Y \approx \sum_{i=1}^{m} \sigma'_i u'_i {v'}_i^T S_0^{-1} X \ .$$

The compression error $L = ||(WS_0 - \sum_{i=1}^{m} \sigma'_i u'_i {v'}_i^T)S_0^{-1}X||_F$ is demonstrated below:

$$
\begin{aligned}
L^2 &= ||(WS_0 - \sum_{i=1}^{m} \sigma'_i u'_i {v'}_i^T)S_0^{-1}X||_F^2 \\
&= \left\| \sum_{i=m+1}^{r} \sigma'_i u'_i {v'}_i^T S_0^{-1} X \right\|_F^2 \\
&= \sum_{j=m+1}^{r} \sum_{i=m+1}^{r} \sigma'_i \sigma'_j \mathbf{Trace}(u'_i {v'}_i^T X X^T v'_j {u'}_j^T) \\
&= \sum_{j=m+1}^{r} \sum_{i=m+1}^{r} \sigma'_i \sigma'_j \mathbf{Trace}({u'}_j^T u'_i {v'}_i^T S_0^{-1} X X^T (S_0^{-1})^T v'_j) \\
&= \sum_{i=m+1}^{r} {\sigma'}_i^2 \mathbf{Trace}({v'}_i^T S_0^{-1} X X^T (S_0^{-1})^T v'_i) \\
&= \sum_{i=m+1}^{r} {\sigma'}_i^2 ||{v'}_i^T S_0^{-1} X||_F^2 \ ,
\end{aligned}
$$

which is still a complex function that involves the activation $X$, the diagonal matrix $S_0$, the singular vector $v'_i$ and the singular value $\sigma'_i$. As a result, compression error is not directly related to the singular value, and the conventional SVD compression by truncating the smallest singular values may lead to suboptimal compression error.

## A.3 COMPRESSION LOSS OF SVD-LLM

In SVD-LLM, we also formulate the linear layer as $Y = (WS)S^{-1}X$, where $S^{-1}XX^T (S^{-1})^T = I$. The compression is made by keeping the largest $m$ out of total $r$ singular values of $WS$. The compression loss $L$ is demonstrated as:

$$L^2 = \|WX - W'X\|_F^2 = \|WSS^{-1}X - \mathbf{SVD}(WS)S^{-1}X\|_F^2$$

$$= \|(WS - \mathbf{SVD}(WS))S^{-1}X\|_F^2$$

$$= \left\|\left(WS - \sum_{i=1}^m \sigma_i u_i v_i^T\right)S^{-1}X\right\|_F^2$$

$$= \left\|\sum_{i=m+1}^r \sigma_i u_i v_i^T S^{-1}X\right\|_F^2$$

$$= \sum_{j=m+1}^r \sum_{i=m+1}^r \sigma_i \sigma_j \mathbf{Trace}\left(u_i v_i^T S^{-1} X X^T \left(S^{-1}\right)^T v_j u_j^T\right)$$

$$= \sum_{j=m+1}^r \sum_{i=m+1}^r \sigma_i \sigma_j \mathbf{Trace}\left(u_i v_i^T \left(S^{-1} X X^T \left(S^{-1}\right)^T\right) v_j u_j^T\right)$$

$$= \sum_{j=m+1}^r \sum_{i=m+1}^r \sigma_i \sigma_j \mathbf{Trace}\left(u_i v_i^T v_j u_j^T\right)$$

$$\because v_i^T v_i = u_i^T u_i = 1; v_i^T v_j = u_i^T u_j = 0, \mathbf{Trace}\left(v_i v_i^T\right) = \mathbf{Trace}\left(u_i u_i^T\right) = 1, \forall i \neq j$$

$$\therefore L^2 = \sum_{j=m+1}^r \sum_{i=m+1}^r \sigma_i \sigma_j \mathbf{Trace}\left(u_i v_i^T v_j u_j^T\right) = \sum_{i=m+1}^r \sigma_i^2 \mathbf{Trace}\left(u_i v_i^T v_i u_i^T\right) = \sum_{i=m+1}^r \sigma_i^2$$

Therefore, the squared loss $L^2$ is equal to the sum of the squared singular values. Therefore, truncating the smallest singular values achieves the lowest compression loss.

### A.4 SPECTRUM ANALYSIS OF SINGULAR VALUES DECOMPOSED BY SVD-LLM

In general, SVD is useful for compression when the matrix to be compressed shows a sharp decay of the singular values. Since SVD-LLM decomposes the multiplication of the weight matrix $W$ and its corresponding whitening matrix $S$ instead of the original weight matrix $W$, which is different from the weight decomposition in the previous work (Yuan et al., 2023; Hsu et al., 2022), to study whether SVD compression is also applicable in SVD-LLM, we select the Query ($W_Q$) and Key ($W_K$) weight matrices and show the spectrum of singular values of their multiplication with corresponding whitening matrices $S_Q$ and $S_K$. As shown in Figure 6, most of the single values are less than or around 100 with only a few extremely large values, indicating that SVD is applicable in SVD-LLM.

### A.5 COMPARISON WITH DRONE

Previous work DRONE (Chen et al., 2021) also proposes their data-aware method for SVD compression. They even provide a theoretical analysis to prove the optimal solution that their method achieves. Specifically, DRONE represents the low-rank compressed weight matrix $W'$ by $WM$. It performs SVD on both weight matrix $W = U_w S_w V_w^T$ and the transpose of input activation $X^T = U_x S_x V_x^T$ and then split these decomposed matrices as follows:

$$U_W = \begin{bmatrix} U_{W,r} & \bar{U}_{W,r} \end{bmatrix}, S_W = \begin{bmatrix} S_{W,r} & 0 \\ 0 & 0 \end{bmatrix}, V_W = \begin{bmatrix} V_{W,r} & \bar{V}_{W,r} \end{bmatrix}$$

$$U_X = \begin{bmatrix} U_{X,t} & \bar{U}_{X,t} \end{bmatrix}, S_X = \begin{bmatrix} S_{X,t} & 0 \\ 0 & 0 \end{bmatrix}, V_X = \begin{bmatrix} V_{X,t} & \bar{V}_{X,t} \end{bmatrix}.$$

where $r$ and $k$ are the rank of the $W$ and $X$. $U_{W,r}, V_{W,r}, U_{X,t}, V_{X,t}$ denote corresponding row spaces and column spaces and $\bar{U}_{W,r}, \bar{V}_{W,r}, \bar{U}_{X,t}, \bar{V}_{X,t}$ are null spaces. Through theoretical deduction,

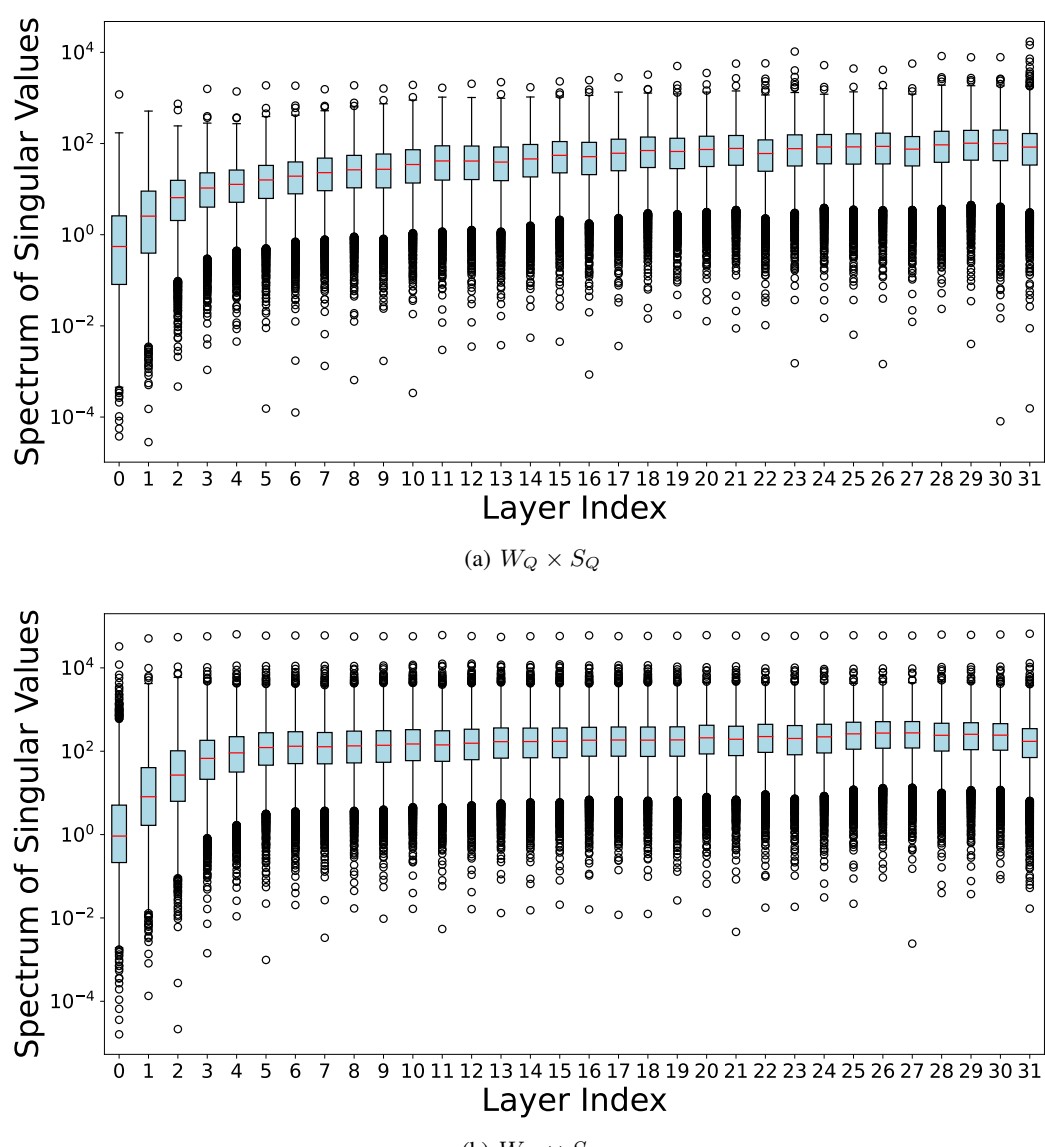

(a) $W_Q \times S_Q$

(b) $W_K \times S_K$

Figure 6: The singular value spectrum of the decomposed matrices across layers in LLaMA-7B.

DRONE converts the minimization of compression loss $||WX - W'X||_F = ||WX - WMX||_F$ to the minimization of $\left\|S_{W,r}V_{W,r}^T V_{X,t}S_{X,t} - S_{W,r}V_{W,r}^T M V_{X,t}S_{X,t}\right\|_F$, whose optimal value $L_{min}$ is the rank-k truncated SVD of $Z = S_{W,r}V_{W,r}^T V_{X,t}S_{X,t}$ by the fundamental property of SVD decomposition. To achieve the optimal value, DRONE formulates a solution $M = V_{W,r}S_{W,r}^{-1}Z_k S_{X,t}^{-1}V_{X,t}^T$, where $Z_k$ is the rank-k SVD truncation of $Z$.

In short, compared with DRONE, `SVD-LLM` is also optimal with the same theoretical compression loss as DRONE. Moreover, `SVD-LLM` has **three** key advantages. Below is our detailed explanation.

**`SVD-LLM` is also optimal with the same theoretical compression loss as DRONE.** The theoretical minimum compression loss $L_{min}$ is the F-norm loss of rank-k SVD truncation of $WX$, which has also been achieved by DRONE in their paper. Unlike DRONE, `SVD-LLM` constructs the whitening matrix $S$ so that $S^{-1}X$ is orthonormal. Therefore, we have $||AS^{-1}X||_F = ||A||_F$. Suppose that we decompose $S$ with SVD to $U_s, S_s, V_s$, we can have $S_s = S_x, U_s = U_x, Us = Ux, V_s = QV_x$, where $Q$ is an orthogonal matrix. The matrix $WS$ to which `SVD-LLM` applies SVD could be represented

Table 9: Compression loss of the randomly generated weight and activation matrices with different shapes under 50% compression ratio using `SVD-LLM`, Drone, and the theoretical minimum.

| Loss | $[128 \times 128] \times [128 \times 128]$ | $[2048 \times 2048] \times [2048 \times 2048]$ | $[4096 \times 4096] \times [4096 \times 4096]$ |
|---|---|---|---|
| Minimum | 276.1130 | 17784.2637 | 50321.9141 |
| Drone | 276.1130 | 17785.6992 | 50337.2148 |
| SVD-LLM | 276.1130 | 17784.2676 | 50321.9727 |

Table 10: Compression Time of the randomly generated weight and activation matrices with different shapes using `SVD-LLM` and Drone. The compression time is measured for 5 times' compression.

| Time | $[128 \times 128] \times [128 \times 128]$ | $[2048 \times 2048] \times [2048 \times 2048]$ | $[4096 \times 4096] \times [4096 \times 4096]$ |
|---|---|---|---|
| Drone | 0.07 seconds | 5.81 seconds | 30.35 seconds |
| SVD-LLM | 0.02 seconds | 1.98 seconds | 10.37 seconds |

by $U_w S_w V_w^T U_s S_s V_s^T$. Suppose that we use **Trunc.**$(C)$ to represent the rank-k truncation of the matrix $C$ during SVD compression, the compression loss $L$ is derived as follows:

$$L = ||WX - W'X||_F = ||(WSS^{-1}X - \mathbf{SVD}(WS)S^{-1}X)||_F = ||(WS - SVD(WS))S^{-1}X||_F$$

$$= ||\mathbf{Trunc.}(WS)S^{-1}X||_F = ||\mathbf{Trunc.}(WS)||_F$$

$$= ||\mathbf{Trunc.}(U_w S_w V_w^T U_s S_s V_s^T)||_F = ||\mathbf{Trunc.}(WXQ^T)||_F = L_{min}$$

Therefore, `SVD-LLM` shares the same theoretical compression loss as DRONE.

**Advantage #1: DRONE incurs out-of-memory when compressing LLMs due to its requirement of storing the full large-scale activations, whereas `SVD-LLM` is feasible.** To achieve data-awareness during compression, DRONE caches all input activations $X$ and spans them to calculate the corresponding singular vectors and singular values. In the DRONE paper, the authors apply DRONE to small LMs such as BERT. However, the activations generated by LLMs are often extremely large and are much larger than BERT. For example, to compress LLaMA-7B with 5,000 calibration data by DRONE, the total memory to cache the activation for a single weight matrix at a time is 5000 (data number) $\times$ 256 (seq_len) $\times$ 11008 (dim) $\times$ 32 (fp32) $\div$ 1024 $\div$ 1024 $\div$ 1024 = 419GB, which is more than 5 times larger than the memory provided by the NVIDIA A100 GPU, which has 80GB memory. Therefore, applying DRONE for LLM compression is infeasbile.

In contrast, `SVD-LLM` incrementally updates its $XX^T$ matrix by adding the $xx^T$ of each new input $x$. As such, `SVD-LLM` eliminates the need to store the full activations, which requires only the storage of the matrix $XX^T$, which is considerably smaller than the full input activation. To compress LLaMA-7B with 5,000 calibration data, `SVD-LLM` requires only 11008 $\times$ 11008 $\times$ 32 $\div$ 1024 $\div$ 1024 $\div$ 1024 = 3.6GB. Compared to DRONE, `SVD-LLM` achieves 116.38 times less memory reduction than DRONE. In our paper, we use WikiText-2 as a dataset. If we use DRONE, the total memory to cache the activation of a single weight matrix is larger than 24,600GB. In contrast, `SVD-LLM` still requires 3.6GB of memory, which is more than 6,000 times less than DRONE. Due to this advantage, `SVD-LLM` is much more practical to compress LLMs of size 7B or larger compared to DRONE.

**Advantage #2: `SVD-LLM` incurs much shorter compression time compared to DRONE.** DRONE involves more complex matrix operations, leading to longer compression time compared to `SVD-LLM`. To illustrate this, we measured the time required by DRONE and `SVD-LLM` to compress randomly generated weight and activation matrices of varying shapes under 50% compression ratio. The results show that `SVD-LLM` is approximately three times faster than DRONE.

**Advantage #3: `SVD-LLM` has better numerical stability, which leads to superior empirical compression loss.** While `SVD-LLM` shares the same theoretical compression loss as DRONE, DRONE's higher complexity—stemming from additional SVD operations and inverse calculations on large-scale matrices—makes it less numerically stable compared to `SVD-LLM`. This often results in higher empirical compression losses. To illustrate this, we compare `SVD-LLM` and DRONE in terms of empirical compression losses for randomly generated matrices of various shapes. We also include the theoretical minimum value, represented by the rank-k Frobenius norm loss of $WX$. The results are summarized in the following table. As shown, we observe that `SVD-LLM` achieves lower empirical compression losses than DRONE, underscoring its superior numerical stability.

Table 11: Perplexity (↓) of `SVD-LLM` and FLAP on WikiText-2 to compress LLaMA-7B under different compression ratios. The better performance is marked in bold. The relative performance gain of `SVD-LLM` compared to FLAP is marked in green inside bracket.

| RATIO (MEM.) | 20% (10.2GB) | 40% (7.76GB) | 60% (5.35GB) | 80% (2.58GB) |
|---|---|---|---|---|
| FLAP | 7.99 | 14.43 | 106.87 | 15023 |
| SVD-LLM | **7.73** (↓3%) | **9.27** (↓36%) | **15.00** (↓86%) | **31.79** (↓99%) |

Table 12: Comparison of LLaMA-3B (compressed from LLaMA-7B by `SVD-LLM`) and original StableLM-3B (Tow et al.) trained from scratch. Both the throughput and the peak memory footprint during the inference are measured with batch size=32, sequence length = 128 on single A100 GPU.

| MODEL | Throughput | Peak Mem. | Openb. | Arc_e | WinoG. | HellaS. | PIQA | MathQA | Average↑ | TruthfulQA↑ | GSM8K↑ |
|---|---|---|---|---|---|---|---|---|---|---|---|
| StableLM-3B | 8463 Tokens/sec | 9.41 GB | 0.19 | 0.51 | 0.55 | 0.47 | 0.69 | 0.21 | 0.44 | 0.22 | 0.02 |
| LLaMA-3B | 9254 Tokens/sec | 7.43 GB | **0.27** | **0.54** | **0.60** | **0.49** | 0.68 | 0.19 | **0.46** (↑5%) | **0.23** (+ 0.01) | **0.04** (+ 0.02) |

## A.6 COMPARISON WITH FLAP

Recent work FLAP (An et al., 2024) is also a post-training structured-pruning method. Below we compare the perplexity of `SVD-LLM` and FLAP on WikiText-2 under different compression ratios when compressing LLaMA-7B. As shown in Table 11, `SVD-LLM` consistently outperforms FLAP, especially under high compression ratios.

## A.7 COMPARISON WITH SMALLER LLMS PRE-TRAINED FROM SCRATCH

To compare the performance between `SVD-LLM` and scratch training, following the previous experimental design (Ma et al., 2023), we compress LLaMA-7B to the size of the 3B parameter with `SVD-LLM` and select StableLM-3B (Tow et al.) as the baseline for comparison. As shown in Table 12. LLaMA-3B compressed from LLaMA-7B by `SVD-LLM` achieves better accuracy in all datasets, indicating that `SVD-LLM` could even achieve better accuracy than some scratch training methods. Furthermore, `SVD-LLM` ensures higher throughput and lower memory consumption than StableLM-3B as shown in the table, which also meets our efficiency analysis and discussion in Section 4.2.

Table 13: Compression time of `SVD-LLM` and ASVD on LLaMA-7B under 20% compression ratio. The relative speedup is marked in green color inside bracket.

| SVD-LLM | | | ASVD | | |
|---|---|---|---|---|---|
| Truncation-Aware Data Whitening | Parameter Update with Sequential Low-rank Approximation | **Total** | Normalize | Search | **Total** |
| 10min | 3.5h | **3.5h** (↓36%) | 5min | 5.5h | **5.5h** |

## A.8 COMPRESSION SPEED EVALUATION

In addition to compression performance, we also evaluate the compression speed of `SVD-LLM` and the baselines. Specifically, we measured the GPU hours used for `SVD-LLM` and ASVD when compressing LLaMA-7B under 20% compression ratio on an A100 GPU. The results are shown in Table 13. As shown, ASVD takes about 5.5 hours, while `SVD-LLM` completes the compression process in 3.5 hours, which is 36% times faster. When breaking down the time, most of the time consumed by ASVD is dedicated to searching for the specific compression ratio for each weight matrix based on its calculated importance score. In contrast, `SVD-LLM` maintains a consistent compression ratio in all weight matrices and thus eliminates the time-consuming search process.

## A.9 Contents Generated from LLMs Compressed by SVD-LLM and ASVD

Some examples of sentences generated by LLaMA-7B compressed with `SVD-LLM` and ASVD are shown in Table 14. The sentences generated by the model compressed by `SVD-LLM` exhibit better fluency, relevance, and informativeness compared to those compressed by ASVD. More importantly, when the compression ratio is increased to 40%, the previous state-of-the-art method ASVD completely loses its generation ability. In contrast, even when the compression ratio is up to 80%, `SVD-LLM` is still capable of generating complete sentences.

Table 14: An example of contents generated by the fine-tuned LLaMA-7B with LoRA after being compressed by `SVD-LLM` and ASVD under different compression ratios. The input is marked in bold and the normal texts are the generated sentences.

| Ratio | SVD-LLM | ASVD |
|---|---|---|
| Original | **What is the responsibility of the AI assistant?** The responsibility of an AI assistant is to complete the job without any hesitation. Also, the AI assistant has no problem with taking a decision and executing it. An AI assistant is responsible to complete the task and it needs to learn from his past experience to take the best possible decision without any hesitation... | |
| 20% | **What is the responsibility of the AI assistant?** The AI assistant's role and responsibilities are often ambiguous and can vary depending on the context and the type of assistant being used. Some AI assistants may have preset responses to common queries, but their behavior can also depend on the user's interaction and expectations... | **What is the responsibility of the AI assistant?** 1. To act in the same way as the programmer intended. 2. To make a value assessment of the user's intentions and offer an appropriate response. 3. To help the user carry out specific tasks or follow a strict script. 4. To make the system as independent of the user as possible... |
| 40% | **What is the responsibility of the AI assistant?** 1. The AI assistant is responsible for performing tasks that can only be done by AI technology such as scheduling appointments, setting reminders and making suggestions, while also providing quick and accurate responses. 2. The AI assistant is also responsible for learning and updating its knowledge base through continuous improvement, thus making it more efficient and effective. 3. The AI assistant is responsible for handling and managing user-generated and customer-oriented input... | **What is the responsibility of the AI assistant?** 1. Define the definition of the job specification. 2. Define the role of the AI assistant for the job. 3. Define the skills necessary in order to carryout the tasks assigned to the job to do the work of the same capacity in the market. 4. Determine the evaluation of the job competence of the AI assistant to ensure their professional integrity. 5. What type of AI assistant jobs are available? 6. What industries require technology assistance work to complete the tasks that it own personnel cannot perform... |
| 60% | **What is the responsibility of the AI assistant?** In a nutshell, the responsibility of the AI assistant could vary depending on the task, but generally, the focus would be on automatic tasks, without the need for human intervention. Some common tasks could include setting reminders, scheduling appointments, and making routine household or productivity tasks. The AI assistant also serves as a backup or a relief system, taking on responsibilities when the user is not available ... | **What is the responsibility of the AI assistant?** 2.3. ?? Brush for a discussion I wonder is it worth doing is important.2- It isn't useful just reducing labor costs; it helps employees feel a sense of connected to your attention which leads to better workplace values among staffers and leads to long relationships among org... |
| 80% | **What is the responsibility of the AI assistant?** Our Design is based on our understanding of the world, and we are actively learning, adapting and adapting, so we're always evolving new ideas, which we see to be most unique and relevant in our community... | **What is the responsibility of the AI assistant?** ygua AIeltemperaturen/2, (64mbz/.3/.1/, 7.kbld.org.0/2/ In these puthebout les bnvols n merginels ... |

## A.10 More Ablation Studies

**`SVD-LLM` + Normal LoRA Fine-tuning v.s. `SVD-LLM` + Sequential LoRA fine-tuning.** To illustrate the superiority of the designed parameter update with the sequential low-rank approximation in `SVD-LLM`, which is a kind of sequential LoRA fine-tuning strategy over the normal LoRA fine-tuning strategy, we compare the compression performance of `SVD-LLM` by applying either of these two fine-tuning strategies. Let's denote `SVD-LLM` (SFT) as `SVD-LLM` by applying sequential LoRA fine-tuning and `SVD-LLM` (NFT) as `SVD-LLM` by applying normal LoRA fine-tuning. As shown in Table 15, `SVD-LLM` (SFT) consistently outperforms `SVD-LLM` (NFT), which also reaffirms our analysis in Section 3.2 that optimizing both low-rank matrices $W_u, W_v$ at the same time is not stable and may lead to poor fine-tuning performance.

**ASVD + Sequential LoRA Fine-tuning v.s. `SVD-LLM` + Sequential LoRA Fine-tuning.** Although the designed sequential LoRA fine-tuning strategy could also be applied in other SVD-based LLM compression methods, other methods' performance is still poorer than `SVD-LLM` even when integrated with this strategy for enhancement. To illustrate this, we compare the performance of the

Table 15: Perplexity (↓) of `SVD-LLM` with original LoRA fine-tuning (denoted as `SVD-LLM` (SFT)), ASVD with sequential LoRA fine-tuning (denoted as ASVD (SFT)), and `SVD-LLM` with sequential LoRA fine-tuning (denoted as `SVD-LLM` (SFT)) on WikiText-2 to compress LLaMA-7B under different compression ratios.

| Ratio (Mem.) | 20% (10.2GB) | 40% (7.76GB) | 60% (5.35GB) | 80% (2.58GB) |
|---|---|---|---|---|
| SVD-LLM (NFT) | 7.87 | 11.98 | 16.30 | 80.23 |
| ASVD (SFT) | 8.37 | 14.86 | 44.81 | 271 |
| SVD-LLM (SFT) | **7.73** | **9.27** | **15.00** | **31.79** |

previous state-of-the-art method ASVD when applied with the sequential LoRA finetuning with `SVD-LLM`. Let's denote `SVD-LLM` (SFT) as `SVD-LLM` by applying sequential LoRA fine-tuning and ASVD (SFT) as ASVD by applying sequential LoRA fine-tuning. As shown in Table 15, `SVD-LLM` (SFT) consistently outperforms ASVD (SFT) under various compression ratios.

## A.11 Limitations

There are three limitations of `SVD-LLM`, which are left for future work.

**(1) The compression accuracy still needs to be improved under the high compression ratio.** Although `SVD-LLM` has achieved the state-of-the-art performance compared to previous works such as FWSVD and ASVD, its compression accuracy still suffers from degradation, especially under the high compression ratio. To enhance the practicability of `SVD-LLM` in real-world scenario, its accuracy should be at least comparable to that of the quantization method, including both low-bit and high-bit quantization, rather than being combined with the quantization methods for usage.

**(2) The latency of `SVD-LLM` needs to be optimized while being used to compress the KV cache.** As discussed in Section 4.2, `SVD-LLM` can also be applied to compress the KV cache for memory saving during inference. However, this benefit does not come free. In fact, due to the additional calculation caused by recovering the original key and value states, the inference speed will be impacted by compressing the KV cache with `SVD-LLM`, which should be optimized in the future.

**(3) `SVD-LLM` should be better guided for high-quality generation.** Although `SVD-LLM` achieves low perplexity, as demonstrated in Table 1, it is still possible to generate low-quality content, such as repeated words even under low compression ratios. This phenomenon of compressed LLM has also been mentioned in previous work (Ma et al., 2023) and should be eliminated in the future by guiding the compressed LLM for outputting high-quality generation.

