# OpenReview forum: "SVD-LLM: Truncation-aware Singular Value Decomposition for Large Language Model Compression"
_ICLR.cc/2025/Conference — ICLR 2025 Poster_

### Official Review · Reviewer_kQW6 · 2024-10-23

**Soundness:** 4
**Presentation:** 4
**Contribution:** 4
**Rating:** 8
**Confidence:** 4

**Summary:**

This paper proposes SVD-LLM, a novel SVD-based LLM compression method that addresses efficiency issues in previous methods. It establishes a direct mapping between singular values and compression loss with theoretic proof. Extensive experiments demonstrate the effectiveness of the proposed method and support the author’s claims.

**Strengths:**

1. SVD-LLM can achieve state-of-the-art in terms of compression efficiency and performance on downstream tasks.
2. The inference speedup that SVD-LLM can achieve is inspiring given SVD-LLM’s good performance on PPL and downstream tasks.
3. The research is well presented and it is easy to follow.

**Weaknesses:**

(a) “SVD for LLM Compression” part in “Related works” is insufficient, and it can be improved by discussing more papers related to SVD-based LLM compression and papers that relate to the Atomic Feature Mimicking (AFM) technique, where the latter is also dedicated to model decomposition for LLM compression. Papers that authors should be aware of are listed as follows.

1. Compressing Transformers: Features Are Low-Rank, but Weights Are Not!, AAAI’23

2. LoSparse: Structured Compression of Large Language Models based on Low-Rank and Sparse Approximation, ICML’23

3. Adaptive Rank Selections for Low-Rank Approximation of Language Models, NAACL’24

(b) The experiments on the latest model are missing. Authors should test their proposed method, i.e., SVD-LLM, and compare it with baselines on the latest model such as LLaMA-3-8B.

**Questions:**

The concept of “SVD-LLM (U)” in S4.3 - "Modular Sensitivity Study" needs to be clarified. From what I understand, the core part of SVD-LLM is the truncation-aware data whitening, the concept of “SVD-LLM (U)” thus needs to be clarified since SVD-LLM will be degraded to normal truncated-SVD when "truncation-aware data whitening" is disabled.

---

### Official Review · Reviewer_8cQQ · 2024-10-28

**Soundness:** 3
**Presentation:** 2
**Contribution:** 3
**Rating:** 5
**Confidence:** 4

**Summary:**

This paper introduces a LLM compression approach based on SVD called SVD-LLM. SVD-LLM incorporates a whitening process to maintain the orthogonality of activations and align the number of retained singular values and the compression loss. After compressed via SVD, SVD-LLM further finetune the model for better performance.

**Strengths:**

1. The activation whitening is supported by theoretical analysis, which is convincing.
2. The experiments are extensive, demonstrating improvements comparing to selected baselines.
3. SVD-LLM delivers practical benefits in terms of inference speedup and memory reduction on hardware.

**Weaknesses:**

1. The comparisons in the main experiments might be \textbf{unfair}, as SVD-LLM adds extra finetuning additional finetuning that is not applied to other SVD-based baselines, which could drastically improve performance under high compression ratios. Extra finetuning is not a novel technique and can be seamlessly integrated into the baselines.
2. Theorem 3.2 and Corollary 3.3 only hold when $S$ is calculated on the fly, but in practice SVD-LLM precomputes $S$ using a calibration set, and it is unclear if the orthogonality of activations  is preserved with this approach. Further experiments are needed here to address this gap, for example, the orthogonality of whitened activations using the precomputed $S$.
3. The paper lacks validation of the method on larger models, such as those at the 70B scale, which is crucial for evaluating the feasibility of model compression approaches.

**Questions:**

1. The paper claims to use LoRA finetuning in Section 3.2 but it is not clear how it is done. Does this introduces additional parameters, or fused afterward? Or the finetuning is directly applied on $W_u$ and $W_v$?
2. Section 4.2, the authors discuss memory reduction for the KV cache at the expense of recomputation overhead. How does this impact throughput, and by how much?
3. In Section 4.4 and Table 8, SVD-LLM is combined with QuIP#. More details are needed regarding how this combination is executed.
4. In Algorithm 3,  there is a layer-wise closed-form update procedure that was not mentioned in the paper.

---

> ### Author Response · Authors · 2024-12-03
> **Only 1 day left for discussion. We have added one more experiment and are looking forward for discussion.**
>
> Dear Reviewer 8cQQ,
>
> We sincerely appreciate your valuable comments and suggestions on our paper. We have made every effort to address your concerns and hope these updates align with your expectations.
>
> Although we have not yet received further responses, we remain motivated and are continuously working to improve our paper. Specifically, we compared **SVD-LLM\*—SVD-LLM without the proposed sequential LoRA fine-tuning**—with current pruning-based LLM compression methods. As demonstrated in the results, SVD-LLM continues to outperform pruning-based methods even without LoRA. We promise to include this analysis in our new revised version upon acceptance. We believe the current set of experiments is sufficiently fair, as it compares SVD-LLM against both SVD-based and pruning-based methods, with and without LoRA.
> | METHOD      | 10 GB | 9 GB  | 8 GB  | 7 GB  |
> |-------------|-------|-------|-------|-------|
> | LLM-Pruner  | 9.88  | 12.21 | 18.94 | 21.68 |
> | SliceGPT    | 8.78  | 12.73 | 16.39 | 21.68 |
> | BlockPruner | 9.4   | 12.76 | 19.78 | 43.05 |
> | FLAP        | 8.15  | 11.24 | 14.01 | 19.83 |
> | SVD-LLM*    | 8.01  | 10.26 | 13.68 | 17.24 |
> | SVD-LLM     | 6.92  | 7.38  | 8.02  | 9.23  |
>
> We also deeply appreciate your acknowledgement that `Thanks to the authors for addressing all questions.` in the last response. Your recognition means a great deal to us. **Since there are fewer than 24 hours remaining in the discussion period, we kindly ask if you have any additional concerns. We would be grateful for the opportunity to address them promptly and hope this might lead you to reconsider your score.**
>
> Thank you very much!
>
> Best wishes,
>
> All authors of Submission 2301

---

### Official Review · Reviewer_P8yJ · 2024-10-31

**Soundness:** 3
**Presentation:** 4
**Contribution:** 3
**Rating:** 6
**Confidence:** 4

**Summary:**

With the remarkable capability of large language models (LLMs) in the field of natural language processing, their large model size becomes a major obstacle for practical deployment. To this end, the paper proposes a Singular Value Decomposition (SVD)-based LLM compression method, which aims to address two key limitations faced by existing SVD methods in compression: the truncation of small singular values that may lead to higher compression loss and the lack of model parameter updating after SVD truncation.

**Strengths:**

- The paper introduces SVD-LLM, a novel SVD-based compression method that incorporates a truncation-aware data whitening strategy to directly map singular values to compression loss, allowing for more informed truncation decisions.
- The paper provides a rigorous mathematical foundation for the proposed truncation-aware data whitening technique, supported by theoretical proofs and lemmas.
- It proposes a parameter update mechanism with sequential low-rank approximation to compensate for accuracy degradation after compression.
- The authors evaluate SVD-LLM across various datasets and models from three different LLM families at various scales, demonstrating its superiority over existing SVD-based methods, especially at high compression ratios.

**Weaknesses:**

- The authors should evaluate SVD-LLM on the latest models in LLaMA family, such LLaMA-3-8B.
- Authors are recommended to analyze the computation costs of SVD-LLM and provide the evaluation of its time expense.
- No comparison with the latest SOTA methods, such as FLAP: Fluctuation-based Adaptive Structured Pruning for Large Language Models. Yongqi An, et al. AAAI 2024, available on arXiv since Dec. 2023.

**Questions:**

Please refer to the weaknesses.

---

### Official Review · Reviewer_3PXg · 2024-11-12

**Soundness:** 3
**Presentation:** 3
**Contribution:** 2
**Rating:** 6
**Confidence:** 4

**Summary:**

The authors investigate the compression of large language models through low rank approximation. They point out that the existing works in this direction have the following two problems. First, existing works approximate the weight matrix without considering the input activation matrix. Second, existing works do not finetune the model after the truncation. Consequently, they propose SVD-LLM, which addresses both issues by data whitening and LoRA finetuning. Their experiments show that their method not only outperforms other SVD approximation methods, but also outperforms other compression methods like structured pruning or model quantization. Furthermore, real inference speedup and memory saving can be observed without specialized hardware.

**Strengths:**

1. The writing is clear and easy to follow.
2. The empirical comparison is extensive. The authors not only compared with the SVD-based baselines, but also compared with other compression methods like structured pruning and quantization to demonstrate the superiority of the proposed method.
3. Real inference speedup and memory saving can be observed without specialized hardware. Unlike some unstructured pruning and/or quantization methods, which require specialized hardware for speedup, this makes the proposed method more practical and applicable.

**Weaknesses:**

1. As mentioned by SAC, the concept of data aware compression is not new and already exists in [1]. Additionally, in [1] they claim optimality for the problem considered, while this paper seems to offer weaker guarantees. (From my understanding, this paper shows that given the basis/decomposition of their choice, pruning the smallest entries of the middle diagonal matrix is optimal. On the other hand, [1] claim that both their choice of the basis and the truncation is optimal)
2. The LoRA training part of this paper also seems not to be novel as existing in LLM-pruner. In addition, the claim for the necessity of sequential finetuning for W_u, W_v seems odd. For lora finetuning of transformers, similar structure exists for key and query attention matrix W_k, W_q, but existing literature does not report any issues for fine tuning.


[1] Chen, P., Yu, H. F., Dhillon, I., & Hsieh, C. J. (2021). Drone: Data-aware low-rank compression for large nlp models. Advances in neural information processing systems, 34, 29321-29334.

**Questions:**

1. Can you clarify the difference/advantage compared to Drone? Specifically, are the methods equivalent (It seems not so as I do not see a simple connection between Cholesky and eigendecomposition)? If not, can you show that your method is also optimal?
I acknowledge that the experiment is done at a different scale (Bert vs LLM), so verifying this direction works at a larger scale itself is a contribution, but seems not enough for a full paper.
2. Can you show the performance or LoRA fine tuning without the sequential finetuning process proposed by the paper? This can be one of the novelties of the paper but requires careful verification. The failure of original finetuning might be because of bad choice of hyperparameters for training.

---

> ### Comment · Reviewer_3PXg · 2024-11-28
>
> Hi,
>
> Let's be clear: I am still open to adjusting my score. It’s just that, given there is still roughly a week left, more discussion would give me greater confidence in my decision.
>
> To convince me, I suggest that the authors address the following.
> 1. Argue the novelty/contribution from areas other than the optimal compression part.
> 2. Acknowledging more the work of DRONE, which to me is still quite conceptually similar. Whether improving the computation/storage is a significant contribution is subjective. Calling this an improvement is fine, but like stating that DRONE is unusable as in Sec 3.1 is a bit too much.
>
> I think the problem is actually point 1, but we are discussing point 2 because of the lack of a strong argument for point 1.
> We can still do the discussion on 2 though.
>
> > SVD-LLM is NOT just fixing some edge cases of DRONE.
>
> I don't know why you take my sentence out of context.
>
> So the thing to be discussed here is whether SVD-LLM and DRONE will always output the same thing, disregarding time complexity, memory usage, and numerical error.
>
> You claim that there is a case where outputs are different and the output of SVD-LLM is better.
> I questioned that and asked for the significance of the case, but you did not respond to me.
>
> If you can prove that SVD-LLM and DRONE output differently for significance cases and the output of SVD-LLM is better, then I am convinced, and you probably want to tell the author's of DRONE to fix their paper and do some kind of clarification as they seem to claim optimality.
>
> Otherwise, I think of this as time complexity/memory usage/numerical error optimization of more or less the same conceptual algorithm, where the degree of contribution is subjective.
>
> > DRONE can NOT even compress a 7B LLM using a NVIDIA A100 GPU with 80GB memory.
>
> I feel like this statement should be quantitative instead of qualitative, so like on what dataset and what is your memory estimation of DRONE. For example, openbookqa only has roughly 5000 samples, which DRONE should probably work. There is larger ones, but it is weird to say DRONE always don't work for LLM.
>
> It is also kind of weird you are arguing applicability to LLM by how you differ in handling the data dimension.
>
> To me, there are many ways to fix the memory usage, but basically you don't have to store them all on GPU.
> You only need the activation X one at a time, so you can just store it else where or you might even able to do the computation on CPU.
> This is not too important though, as I stated this is just computation/memory optimization, where there are contributions but the significance is debatable.
>
> > If the reviewer believes DRONE can compress LLMs by using its original algorithm or some easy fix, please let us know how it can be done.
>
> Sure. The authors claim that the main difference is whether to cache $X$. Let's see whether DRONE really need to do that.
> As we can see, DRONE actually only want $U_X$ and $S_X$, while it is $V_X$ for the data dimension that is causing the memory issue you mentioned.
>
> I believe there should be lots of established way to do this. But one should be storing and eigen decompose $XX^T$ to get $U_X$ and $S_X$. It is kind of similar to what you do in SVD-LLM, but this is pretty standard, so I don't feel like I am copying your solution.
>
> I am slightly tired so I might get this wrong. Actually, I hope I am wrong. Otherwise, this feels easier than I originally thought.
>
> Sincerely yours,
> Reviewer 3PXg

---

> ### Author Response · Authors · 2024-11-30
> **Many thanks for taking time to discuss with us. We really appreciated and are truly grateful.**
>
> Dear reviewer 3PXg,
>
> Thank you for the comments. We believe the discussion is mutually beneficial in the sense that we learn from each other’s perspective, and understand how novelty and contributions can come from different aspects.
>
> Let’s start with your suggested method to help fix DRONE’s applicability issue to compressing LLMs. Basically, what you suggested to improve DRONE is **EXACTLY** what we did in SVD-LLM. In other words, what you suggested is directly borrowing the idea proposed in SVD-LLM to help DRONE. Therefore, we do not think your suggested method is fair to SVD-LLM. What we asked in our previous comment is whether the reviewer could suggest a method that is not borrowed from SVD-LLM.
>
> Next, let’s follow your suggestion on being quantitative, and use the example you used (openbookqa with 5000 samples) to show you quantitative evidence so that we can avoid subjective conjecture such as “DRONE should probably work”. The average sequence length of data in openbookqa is about 256. For LLaMA-7B, the maximum dim of input activation is its intermediate_size = 11,008. Therefore, if we use DRONE, the total memory for caching the activation X for a single weight matrix at a time, as suggested by the reviewer, is 5000 (data number) x 256 (seq_len) x 11008 (dim) x 32 (fp32) / 1024/1024/1024 = **419GB**, which is **more than 5 times larger than the memory provided by NVIDIA A100 GPU**, which has 80GB memory. In contrast, SVD-LLM only requires 11008x11008x32/1024/1024/1024 = **3.6GB.** Therefore, compared to DRONE, SVD-LLM achieves a memory reduction of 419GB/3.6GB, which is **116.38 times less than DRONE.** In our paper, we used WikiText-2 as  the dataset. If we use DRONE, the total memory to cache the activation of a single weight matrix is larger than **24,600GB**. In contrast, SVD-LLM still requires **3.6GB** memory, which is over **6,000 times less than DRONE**. These memory reduction improvements are quantitative and objective; **it is simply NOT fair to count these quantitative and objective contributions as subjective contributions claimed by the reviewer.**
>
>
> Lastly, the “optimal compression part” you referred to in your point 1 should consist of two components: (1) the theoretical optimal output, and (2) the time complexity/memory usage/numerical stability that allows the proposed algorithm to be able to generate the theoretical optimal output. The reason is simple: talking about one while ignoring the other is not fair simply because your real model compression results depend on **BOTH**. DRONE is a great work that contributed to component (1), no doubt about it. **We will follow the reviewer’s suggestion to acknowledge more and revise the statement of DRONE’s contributions in Section 3.1. We really appreciate the reviewer’s suggestion and respect Drone’s contribution.**  Fairly speaking, SVD-LLM achieves the same theoretical minimum as DRONE, and is able to deliver much better model compression results due to its algorithm’s benefits on memory reduction and numerical stability. If you look at the model compression literature, quite a significant number of published papers make contributions to time complexity/memory usage/numerical stability with quantitative and objective results. Therefore, we believe it is **NOT** fair to treat contributions on time complexity/memory usage/numerical stability as subjective and weaken its contributions to model compression.
>
> Once again, we are deeply grateful for your timely response and detailed review provided for our paper. Thank you very much!
>
> Best regards,
>
> All authors of Submission 2301

---

> > ### Comment · Reviewer_3PXg · 2024-12-03
> >
> > I thank the reviewer for the response.
> >
> > I raise my score to 6 under the assumption that the authors will acknowledge more of the contribution of DRONE.
> > I am not confident if this decision is correct. There seems to be no way to guarantee that the authors will deliver the promise.
> > I also still don't fully agree with some of the authors' statement.
> >
> > The following is my response.
> >
> > > Therefore, if we use DRONE, the total memory for caching the activation X for a single weight matrix at a time, as suggested by the reviewer, is 5000 (data number) x 256 (seq_len) x 11008 (dim) x 32 (fp32) / 1024/1024/1024 = 419GB
> >
> > Sure, now I can see that there is a seq_len multiplier so in most cases the naive implementation could not work.
> >
> > >  Basically, what you suggested to improve DRONE is EXACTLY what we did in SVD-LLM. In other words, what you suggested is directly borrowing the idea proposed in SVD-LLM to help DRONE. Therefore, we do not think your suggested method is fair to SVD-LLM.
> >
> > My point to be made here is that some slight changes to DRONE can achieve the same memory usage as SVD-LLM. This is like a less than 5 line change to Algorithm 1 of DRONE.
> >
> > Computational-wise, this changed version of DRONE is still different from SVD-LLM, as they do truncation on different matrices and do different matrix multiplications. The use of $XX^T$ to compute $U$ is also kind of well-known, there are many sources like [1] even though they might not explicitly mention the memory usage benefit.
> >
> > So I think this is a small change (in terms of change in algorithm) with a kind of well-known step that can improve DRONE.
> > Whether the application of this step is easy to think of and whether the contribution is large is subjective.
> >
> > [1] Li, X., Wang, S., & Cai, Y. (2019). Tutorial: Complexity analysis of singular value decomposition and its variants. arXiv preprint arXiv:1906.12085.
> >
> > > If you look at the model compression literature, quite a significant number of published papers make contributions to time complexity/memory usage/numerical stability with quantitative and objective results. Therefore, we believe it is NOT fair to treat contributions on time complexity/memory usage/numerical stability as subjective and weaken its contributions to model compression.
> >
> > I am not saying contributions on time complexity/memory usage/numerical stability are not important.
> > I am saying whether the change is straightforward and the degree of contribution/novelty of the change is subjective.
> >
> > I am not familiar with the entire model compression literature, so I don't know if there is work that focus on improving the time complexity/memory usage/numerical stability of existing methods. But to me, if they propose to compute something that is largely different to improve the time complexity/memory usage, then that is something different to what we are discussing.
> >
> > Sincerely yours,
> > Reviewer 3PXg

---

> ### Author Response · Authors · 2024-12-03
> **Thanks for raising the score, we are truly grateful.**
>
> Dear Reviewer 3PXg,
>
> First, we are deeply grateful to the reviewer for raising the score. Given that the deadline for submitting revision has passed, we are not able to submit another revised version to OpenReview. **However, we will deliver our promise in our next version to acknowledge more of the contribution of DRONE. To do so, here is our specific revision plan:**
>
> 1. We will replace the inaccurate statement in Section 3.1 `"which has been achieved by Drone on small model but unable to be applied for LLM."` with `"Compared with DRONE, SVD-LLM makes improvements in terms of memory, speed, and numerical stability."`
>
> 2. We will include the proof that DRONE could achieve the same optimal compression loss as SVD-LLM for further acknowledgement in Section 3.1.
>
> 3. We will also include the analysis of memory usage of DRONE and SVD-LLM in Section 3.1 for a clearer illustration since memory is the most significant improvement of SVD-LLM over DRONE.
> We kindly remind the reviewer that due to page limit, the detailed comparison between DRONE and SVD-LLM is provided in Appendix A.10.
>
> Second, we want to note that the total number of lines of Algorithm 1 in DRONE is 7. Even including the detailed computation of M, the total number of lines is less than 11. Hence, relatively speaking, changing 5 lines of Algorithm 1 is **NOT** a slight change.
>
> Lastly, our discussions allow us to learn from each other’s perspective on novelty/contributions. We understand and fully respect the reviewer’s perspective. At the same time, as we showed in our previous comment on the memory requirement comparison between DRONE and SVD-LLM, we want to note that **identifying the key bottleneck of existing methods is also important to pushing the state-of-the-arts forward.**
>
> Once again, we are thankful to the reviewer for taking the time and efforts in deeply engaging in the discussion as well as providing suggestions and advice on how our paper writing should be revised. **We promise to release the code of SVD-LLM after the paper’s acceptance to better benefit the community.**
>
> Best wishes,
>
> All authors of Submission 2301

---

### Author Response · Authors · 2024-11-24
**Revision Summary. Hope for further discussion.**

Dear Reviewers,

We sincerely thank you for valuable feedback, which has significantly improved the quality of our paper. Based on your suggestions, we have revised and updated the paper, highlighting changes and additions in **blue** within the revised PDF. Below, we summarize the key revisions made:
- **1. Experiments with one more baseline:** Following suggestion from the reviewer `P8yJ`, we have added the comparison with FLAP[1] in Appendix A.11.
- **2. Discussion on more related works:** Following suggestion from the reviewers `3PXg`, `P8yJ`, and `kQW6`, we have revised the "SVD for LLM Compression” part in “Related works" and included the more related works[1,2,3,4,5]. Following suggestion from the reviewer `3PXg`, we have metioned Drone[5] in Section 3.1 and include the detailed description, including how it works and the comparison with SVD-LLM in Appendix A.10.
- **3. Experiments on more recent and larger-scale LLMs:** Following the suggestions from the reviewers `kQW6` and `P8yJ`, we have conducted the experiments on the latest model LLaMA-3-8B in Appendix A.9. Following the suggestions from the reviewer `8cQQ`, we have added the evaluation on LLaMA-2 70B in Appendix A.9.
- **4. Experiments on Sequential LoRA fine-tuning:** Following the suggestions from the reviewers `3PXg`,  we have added the comparison between the standard LoRA fine-tuning and sequential LoRA fine-tuning in Appendix A.12 and mentioned it in Section 3.2. Following the suggestions from the reviewers `8cQQ` and `P8yJ`,  we have added the comparison of SVD-LLM and previous SoTA method ASVD with sequential LoRA fine-tuning in Appendix A.12 and added the row of SVD-LLM*, which indicates the SVD-LLM without sequential fine-tuning for comparison in Table 1,2,3.
- **5. Experiment on the orthogonality of the whitening matrix $S$:** Following the suggestions from the reviewers `8cQQ`, we have added the orthogonality analysis in Appendix A.12.

After the deadline of revision submission, reviewers also arised two additianal suggestions towards the revision. Although we cannot submit the revised paper on Openreview now, we mark them here and promise to include them in the paper upon paper acceptance.

- **Acknowledge more about Drone in Section 3.1.**

- **Compare pruning-based methods with SVD-LLM without the designed sequential LoRA fine-tuning.**


Once again, we deeply appreciate the time and effort you have dedicated to improving this paper. Your insights have made the paper stronger and more comprehensive. We look forward to any **further discussion or clarification** to better refine our work. Wishing you a Merry Christmas and a Happy New Year!

Bests,

Your friends,

Team of 2301

---
[1] FLAP: Fluctuation-based Adaptive Structured Pruning for Large Language Models. Yongqi An, et al. AAAI 2024

[2] Compressing Transformers: Features Are Low-Rank, but Weights Are Not!, AAAI 2023

[3] LoSparse: Structured Compression of Large Language Models based on Low-Rank and Sparse Approximation, ICML 2023

[4] Adaptive Rank Selections for Low-Rank Approximation of Language Models, NAACL 2024

[5] Drone: Data-aware low-rank compression for large nlp models, NeurIPS, 2021

---

### Comment · Program_Chairs · 2024-11-25
**Comment from Senior Area Chair**

The senior area chair posted the following comment, but the visibility was not set to allow the authors to read it. We are posting it here for transparency.

> I noticed a closely related work (https://proceedings.neurips.cc/paper_files/paper/2021/hash/f56de5ef149cf0aedcc8f4797031e229-Abstract.html) which also conducts low-rank compression by minimizing ||WX - W'X||_F and derives an optimal low-rank solution. Could the authors comment on the relationship and key differences between their approach and this work?

---

> ### Author Response · Authors · 2024-11-25
> **Thanks for the help from PC. SVD-LLM is significantly superior to the mentioned work that was designed for small model comperssion**
>
> We would like to thank the PC for making this comment visible. We also sincerely appreciate the SAC and all reviewers for the engagement during the discussion phase. **In short, SVD-LLM is significantly superior to the work, Drone, that was mentioned by the SAC.** Compared with Drone, SVD-LLM is **also optimal with the same theoretical compression loss**. The detailed proof is provided in the [response](https://openreview.net/forum?id=LNYIUouhdt&noteId=LbdBZuxBbB) to the reviewer `3PXg`. Moreover, SVD-LLM has **four key advantages.** We explain them in detail below.
>
> - **Advantage#1: Only SVD-LLM can perform LLM compression; Drone cannot achieve this.** Drone was designed to compress small models like BERT. It needs to cache all the input activations for the full dataset during compression. As such, **it will cause out-of-memory to compress LLMs like LLaMA-7B even with 80GB GPU memory.**
> In contrast, SVD-LLM incrementally updates its $XX^T$ matrix by adding the $xx^T$ of each new input x. As such, SVD-LLM eliminates the need to store the full activations, requiring only the storage of the $XX^T$ matrix, which is considerably smaller than even a single input activation. Due to this advantage, SVD-LLM is far more practical to compress LLMs of size 7B or larger compared to Drone.
>
> - **Advantage#2: SVD-LLM has better numerical stability, which leads to superior empirical compression loss.**
>  While SVD-LLM shares the same theoretical compression loss as Drone, Drone’s higher complexity—stemming from additional SVD operations and inverse calculations on large-scale matrices—makes it less numerically stable compared to SVD-LLM. This often results in higher empirical compression losses in practice. To illustrate this, we compare SVD-LLM and Drone in terms of the empirical compression losses for randomly generated matrices of various shapes. We also include the theoretical minimum value, represented by the rank-k Frobenius norm loss of $WX$. The results are summarized in the following table. As shown, we observe that SVD-LLM achieves lower empirical compression losses than Drone, underscoring its superior numerical stability.
> | Loss    | [128 x128] x [128 x 128] | [2048 x2048] x [2048 x 2048] | [4096 x 4096] x [4096 x 4096] |
> |---------|--------------------------|------------------------------|-------------------------------|
> | Theoretical Minimum | 276.1130                 | 17784.2637                   | 50321.9141                    |
> | Drone   | **276.1130**                 | 17785.6992                   | 50337.2148                    |
> | SVD-LLM | **276.1130**                 | **17784.2676**                   | **50321.9727**                    |
>
> - **Advantage#3: SVD-LLM incurs much shorter compression time compared to Drone.** Drone involves more complex matrix operations, leading to longer compression time compared to SVD-LLM. To illustrate this, we measured the time required by Drone and SVD-LLM to compress randomly generated weight and activation matrices of varying shapes under a 50% compression ratio. The results show that SVD-LLM is approximately three times faster than Drone.
> | Time    | [128 x128] x [128 x 128] | [2048 x2048] x [2048 x 2048] | [4096 x 4096] x [4096 x 4096] |
> |---------|--------------------------|------------------------------|-------------------------------|
> | Drone   | 0.07 seconds             | 5.81 seconds                 | 30.35 seconds                 |
> | SVD-LLM | **0.02 seconds**             | **1.98 seconds**                 | **10.37 seconds**                 |
>
> - **Advantage#4: The theoretical optimal compression loss cannot always be obtained by Drone’s formula but can always be obtained by SVD-LLM.** This is because Drone formulates the rank-k minimum compression loss as the rank $k$ truncation of $Z = S_{W,r}V^T_{W,r}V_{X,t}S_{X,t}$, where $r,t$ represents the rank of $W, X$, $S_{W,r}, S_{X,t}$ represents the diag matrix with non-zero singular values, $V_{W,r},V_{X,t}$ represents the valid corresponding singular vectors. However, if $k$ is larger than $r$,  the rank-$k$ truncation of $Z$ will incur errors. Therefore, in this case, the theoretical optimal compression loss cannot be obtained by Drone. In contrast, SVD-LLM computes the rank-$k$ truncation of $WS$ with the same shape of $W$, which will not raise errors and hence is able to obtain the theoretical optimal compression loss.
>
> In the **revised sbumission**, we have also discussed Drone in Section 2, mentioned the same theoretical minimum SVD-LLM achieves as Drone in Section 3.1 and provided the detailed illustration in Appendix A.10.
>
> We sincerely thank you for the time and effort you have invested in helping us enhance this paper. We are willing to provide further clarification or additional revisions. Thank you once again for your insightful contributions to our work.
>
> Bests,
>
> Your friends,
>
> Team of 2301

---

### Meta-Review · Area_Chair_PqzD · 2024-12-26

**Metareview:**

This paper proposes a low-rank compression algorithm for compressing a weight matrix in a neural network. The algorithm is truncation aware and is better than naive SVD. Experimental results show promising performance gain over existing low-rank compression algorithms on LLMs.

The main strength of this paper is the strong empirical results, while the main concern from the reviewers is that the optimal low-rank compression derived from this paper is equivalent to the DRONE paper published a few years ago. After extensive discussions between authors and reviewers, the authors agree that the factorization formed in this paper is mathematically identical to DRONE, so the contributions of this paper are (1) Conducting a relative comprehensive experiments on recent LLMs, where DRONE was developed before ChatGPT/Llama so the experiments was mainly conducted on BERT. (2) The authors explicitly show that they can maintain the XX^T matrix in a streaming fashion to make it work for large-scale datasets/models. This was not explicitly mentioned in the DRONE paper. However, after the discussion closed, Reviewer 3PXg found that DRONE mentioned that the XX^T matrix can be maintained in the streaming manner, so this part of novelty becomes unclear.

The paper received 8, 6, 6, 5, and reviewer 3PXg mentioned during discussion to reduce the rating from 6 to 5 due to the point (2) mentioned above, so the actual final rating is 8, 6, 5, 5 which is a borderline paper. Although the novelty may be limited compared to DRONE, given the strong empirical results the AC still votes for weak accept for this paper.

If the paper is accepted, the AC and reviewers would like to see that the authors address point (2) mentioned above in the final version to better compare their work with DRONE.

**Additional Comments On Reviewer Discussion:**

As mentioned above, reviewers all agree with the strengths and the weaknesses of the paper: strength being good empirical results while weakness being the proposed method is mathematically equivalent to DRONE.

---

### Decision · Program_Chairs · 2025-01-22

Accept (Poster)